# An Open-Ended Benchmark and Formal Framework for Adjuvant Research with MLLMs

**Yi Chen**[1,3,6][*], **Yu Zhang**[2,4][*], **Jian Xu**[1,3], **Xu-Yao Zhang**[1,3]✉, **Hua Yue**[2,4]✉, **Xinming Wang**[1,3,6], **Zequan Lyu**[2,5], **Wei Wei**[2,4], **Cheng-Lin Liu**[1,3,6]✉

[1] State Key Laboratory of Multimodal Artificial Intelligence Systems (MAIS)
Institution of Automation, Chinese Academy of Sciences, Beijing, China
[2] State Key Laboratory of Biopharmaceutical Preparation and Delivery
Institute of Process Engineering, Chinese Academy of Sciences, Beijing, China
[3] School of Artificial Intelligence, University of Chinese Academy of Sciences, Beijing, China
[4] School of Chemical Engineering, University of Chinese Academy of Sciences, Beijing, China
[5] School of Advanced Interdisciplinary Sciences, University of Chinese Academy of Sciences, Beijing, China
[6] Zhongguancun Academy, Beijing, China

{yi.chen, xyz, liucl}@nlpr.ia.ac.cn,{jian.xu, wangxinming2024}@ia.ac.cn
{zhangyu21, hyue, weiwei}@ipe.ac.cn,lvzequan24@mails.ucas.ac.cn

## Abstract

Adjuvants play a critical role in modulating immune responses and are central to the development of vaccines and immunotherapies. Yet progress in this field is constrained by data scarcity and incomplete understanding of mechanisms of action, which limit the transition from experience-based design to AI-driven approaches. To address these challenges, we present the first benchmark dedicated to adjuvants, constructed in an open-ended Q&A format and annotated by domain experts. The benchmark comprises 1,294 Q&A pairs and 1,364 formal descriptions, providing a resource for evaluating general-purpose multimodal large language models (MLLMs) and for developing domain-specific systems. We systematically assess 11 closed-source and 19 open-source MLLMs across dimensions including domain-specific Q&A, hallucination rejection, data generation, and instruction following. Results indicate that OpenAI-o1 (STS = 0.7495, LLM Score = 7.7) and DeepSeek-R1 (STS = 0.7415, LLM Score = 7.7) achieved the strongest performance among closed- and open-source models, respectively. In addition, we introduce a formal description framework for representing adjuvant design principles and immune mechanisms as structured abstractions, which can serve as building blocks for future domain-specialized MLLMs. Overall, this work provides a first step toward systematically integrating MLLMs into adjuvant research by offering a dedicated benchmark, comparative evaluation of existing models, and a formal foundation for future development. Data and code will be released at https://github.com/banjiuyufen/Adjuvant-Benchmark.

## 1 Introduction

Artificial intelligence (AI) has become an important driver of scientific discovery, offering new perspectives and tools to address increasingly complex challenges Hessler & Baringhaus (2018); Jumper et al. (2021); Xu et al. (2021); Esteva et al. (2019; 2021); Rajpurkar et al. (2022). Early applications in science often relied on task-specific datasets and bespoke neural architectures Krenn et al. (2020); Wu et al. (2018); Xie & Grossman (2018); de Teresa-Trueba et al. (2023), but recent advances in multimodal large language models (MLLMs) have shifted attention toward more general frameworks capable of integrating heterogeneous information sources Liu et al. (2023); Li et al.

---

[*]Equal contribution.

(2024); OpenAI (2023); Team et al. (2023). These models demonstrate broad capabilities in domains ranging from language and vision to biomedicine, enabling new paradigms for reasoning and analysis He et al. (2024); Xie et al. (2023); Outeiral & Deane (2024); Wang et al. (2025b). Representative work includes LLaVA-Med, a vision-language assistant for biomedical images Li et al. (2023), and BiomedGPT, a generalist biomedical foundation model Zhang et al. (2024b).

Table 1: Cross-domain availability of datasets, methods, and mechanistic principles.

| | Drug Discovery | Protein Structure | Genomics/Omics | Catalyst Design | Battery Materials | **Adjuvants** |
|---|---|---|---|---|---|---|
| Datasets | ✓ | ✓ | ✓ | ✓ | ✓ | ✗ |
| Methods | ✓ | ✓ | ✓ | ✓ | ✓ | ✗ |
| Principles | ✓ | ✓ | ✓ | ✓ | ✓ | ✗ |

Adjuvants are indispensable components of modern vaccines, as they enhance immune responses, prolong protection, and in some cases determine whether a vaccine is clinically viable Glenny et al. (1926); Iwasaki & Omer (2020); Reed et al. (2013). They are particularly critical for emerging infectious diseases and cancer immunotherapy, where rapid and robust immune activation is essential. Despite their importance, the field remains underserved by AI. As shown in Table 1, unlike drug discovery or protein structure prediction—where large-scale datasets and standardized benchmarks already exist—adjuvant research faces three persistent barriers: **(i)** limited systematically curated data, **(ii)** a lack of AI methodologies tailored to adjuvant knowledge, and **(iii)** heterogeneous definitions and mechanisms that complicate systematic modeling Guy (2007). As a result, existing biomedical benchmarks cannot be directly applied, and building domain-specific infrastructure is necessary for progress.

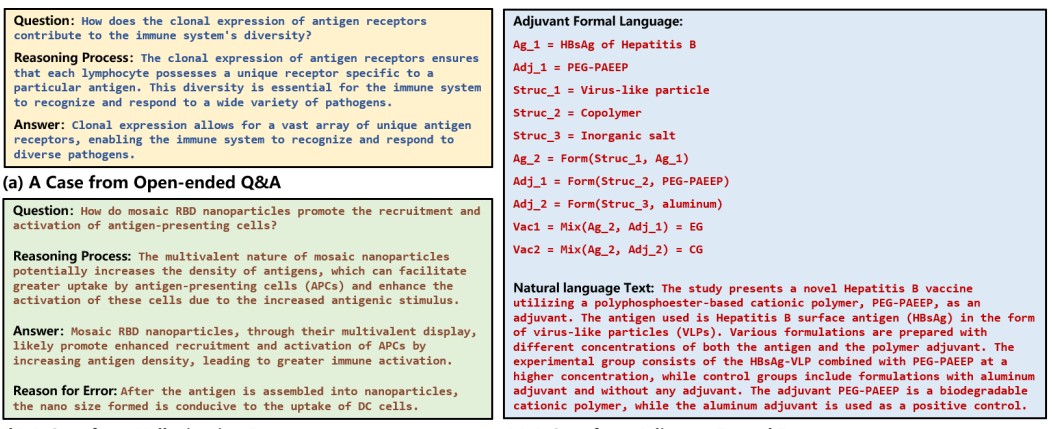

(a) A Case from Open-ended Q&A

(b) A Case from Hallucination Data

(c) A Case from Adjuvant Formal Data

Figure 1: Three Types of Data Display in Adjuvant Benchmark

To address this gap, we present the first benchmark for adjuvants. We adopt an *open-ended Q&A format* to capture mechanistic reasoning, design considerations, and safety issues that cannot be represented through multiple-choice tasks. In parallel, we propose a formal description framework that translates complex biological intuitions into structured abstractions to support reasoning beyond retrieval. Concretely, we generated candidate data with four state-of-the-art MLLMs and conducted rigorous expert annotation across vaccine subdomains. The resulting benchmark consists of three components (Fig. 1): **Open-ended Q&A Data, Hallucination Data**, and **Adjuvant Formal Data**. We then evaluated 11 closed-source and 19 open-source MLLMs on these tasks, assessing domain-specific answering, hallucination rejection, and instruction following, and complemented these with expert-based subjective assessments of generation quality.

Our main contributions are summarized as follows:

- We establish the **first high-quality benchmark dedicated to adjuvants**, explicitly designed to fill a long-standing gap in biomedical AI benchmarks and to support subsequent MLLM research.

- We perform the first systematic evaluation of mainstream general-purpose MLLMs (**11 closed-source and 19 open-source**) on adjuvant knowledge, assessing critical capabilities including data generation, domain-specific QA, hallucination rejection, and prompt follow-

ing. This provides initial conclusions on the capabilities and limitations of current models, and concrete guidance for selecting base models in this domain.

- We introduce **formal descriptions of adjuvants**, converting their complex biological mechanisms into simplified abstractions that can be directly used in training or reasoning. This approach lays the groundwork for future domain-specific MLLMs that combine statistical learning with symbolic reasoning.

## 2 RELATED WORK

### 2.1 ADJUVANTS

Adjuvants are crucial components that are used to improve the effectiveness of vaccines, primarily by stimulating the immune system to improve recognition and response to antigens. By increasing the potency of vaccines, they enable the immune system to respond more rapidly and effectively to pathogens Zhao et al. (2023). Adjuvants can encompass a diverse range of substances, including synthetic small molecule compounds, complex natural extracts, and particulate materials, each contributing uniquely to the modulation of immune responses McKee et al. (2007).

Despite the long-standing and increasing diversity of adjuvants used in vaccines, the mechanisms by which they enhance immune responses are not yet fully understood. With the elucidation of how the innate immune response regulates the adaptive immune response, researchers began to gain insight into the operational mechanisms of adjuvants Coffman et al. (2010). Although this work has provided a certain degree of elucidation regarding the modes of action of adjuvants, a systematic overview and summary of their mechanisms remain scarce due to the broad definition and complex nature of adjuvants.

Recent studies have begun to explore the integration of adjuvants with machine learning to optimize adjuvant selection, such as Nagpal et al. (2018) used the support vector machine (SVM) to develop a hybrid model for predicting A-cell epitopes, which enhances the identification of immune epitopes. Ma et al. (2023) utilized machine learning to identify molecular properties that target Toll-like receptors (TLRs) and designed two new adjuvants to enhance vaccine responses. These effectively promote strong immune responses, significantly suppressing tumor growth and metastases. Chaudhury et al. (2018). used random forest algorithms to develop a predictive model that achieves 92% accuracy in predicting adjuvant conditions based on immune response data, facilitating the identification of immune characteristics of different adjuvants and aiding in the rational pairing of vaccines and adjuvants.

However, these methods often lack generalizability, limiting their effectiveness in complex scenarios. In contrast, MLLMs can learn from vast and diverse modalities, identifying underlying patterns that traditional methods may overlook. This capability enables them to generate more accurate and efficient insights and predictions. By integrating MLLMs with adjuvant research, we aim to accelerate adjuvant development, provide a more responsive approach to public health emergencies, and shift the current paradigm from trial-and-error, experience-based methods to a more AI-driven and efficient process.

### 2.2 SCIENCE BENCHMARK

Recently, there has been increasing attention on MLLMs in scientific research. To evaluate and improve the performance of MLLMs in specific research domains, it is crucial to establish rigorous benchmarks. These not only help in assessing the accuracy and efficiency of the models but also ensure that the evaluation of different methodologies used is consistent and fair across the same studies.

Zhang et al. (2024a) developed ChemBench, an innovative chemical benchmark consisting of 4,100 multiple-choice questions in nine tasks related to chemical molecules and reactions, aiming at objectively measuring the chemical proficiency of large language models (LLMs). Chen et al. (2023) proposed an extensive benchmark study on biomedical text generation, which highlights the strengths and weaknesses of ChatGPT in addressing biomedical tasks, potentially inspiring further advancements in NLP models for biomedical data analysis. Zhang et al. (2025) introduced DataSciBench, a novel and comprehensive benchmarking tool aimed at deeply evaluating the capabilities of LLMs

in data science through natural and challenging tasks. He et al. (2023) proposed a system called SciGuard to control misuse risks associated with AI models in the field of science. They also introduced a red-teaming benchmark, SciMT-Safety, to assess the safety of different systems. Gao et al. (2025) proposed a model-level evaluation framework that emphasizes practical metrics aligned with real-world applications to address the limitations in structure-based drug design (SBDD).

**Summary.** Despite the proliferation of benchmarks in domains like chemistry, biomedicine, and data science, none of them address the unique characteristics of adjuvant research. Existing biomedical benchmarks (e.g., PubMedQA Jin et al. (2019), ChemBench Walker et al. (2010)) mainly evaluate molecular properties, literature summarization, or general biomedical knowledge. In contrast, adjuvants involve heterogeneous substances, multi-scale immune mechanisms, and a lack of structured training data. This makes it impossible to directly apply existing benchmarks to this domain. Our work therefore, fills a critical gap by introducing the first dedicated benchmark for adjuvants, explicitly designed to capture mechanistic reasoning, safety evaluation, and design-oriented knowledge that are absent from prior benchmarks.

# 3 ADJUVANT BENCHMARK

## 3.1 OVERVIEW

Although immunology and adjuvant research have seen significant progress, the systematic integration of MLLMs into this field remains unexplored. To address this gap, we introduce the first benchmark explicitly designed for evaluating MLLMs on adjuvant-related knowledge and reasoning. By curating high-quality academic resources and leveraging multiple state-of-the-art MLLMs, we construct a domain-specific evaluation suite that captures both mechanistic understanding and practical design considerations. The following sections detail the benchmark construction pipeline, expert annotation process, and subsequent analyses of the resulting data.

## 3.2 PIPELINE OF BENCHMARK CONSTRUCTION

The overall construction pipeline is illustrated in Fig. 2. We first collected 739 peer-reviewed papers together with two classic textbooks, from which MLLMs automatically generated approximately $35k$ open-ended Q&A pairs on adjuvants and immunology, each accompanied by an explicit reasoning step (The generation prompts are described in Appendix G.1). To ensure quality and domain relevance, $1.5k$ samples were randomly selected for expert review. After cleaning and careful labeling, $1,294$ high-quality Q&A pairs were retained as the meta dataset. The Krippendorff's $\alpha$ among annotators was 0.8119.

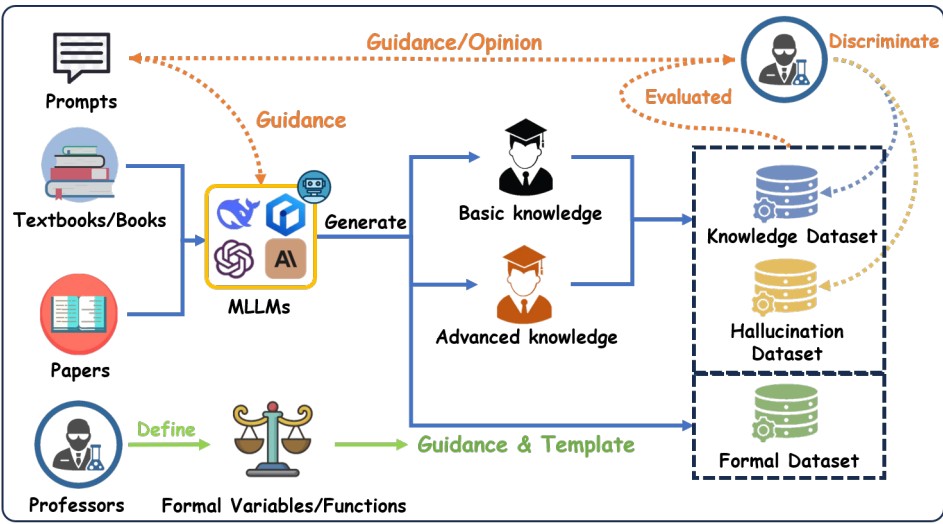

Figure 2: The Benchmark Construction Pipeline.

The annotation team consisted of 13 experts spanning infectious disease, cancer, and bacterial vaccines. All were trained under unified guidelines and evaluated each Q&A–reasoning triplet strictly against the source material. Items were labeled as either valid or hallucinated, with justifications provided for the latter. Detailed preprocessing and annotation workflows are described in Appendix B.

To reduce model-specific bias—particularly since the same system might otherwise generate and answer its own questions—we employed several MLLMs with long-context and multimodal support, including GPT-4o OpenAI, Claude3.5-Sonnet Claude, Ernie4.0-Turbo Baidu, and DeepSeek-R1 Guo et al. (2025).

Following expert annotation, the meta dataset was organized into three complementary components: **Open-ended Q&A Data**, **Hallucination Data**, and **Adjuvant Formal Data**. Each component is described in detail in the subsequent sections.

### 3.3 STATISTICS OF BENCHMARK

#### 3.3.1 OVERALL ANALYSIS

The overall distribution of the benchmark is shown in Fig. 3. Specifically, Fig. 3a illustrates the proportions of different data types, while Fig. 3b summarizes the contributions of various models to Q&A generation. The benchmark is primarily composed of open-ended Q&A items and formal adjuvant data, with GPT-4o and DeepSeek-R1 contributing the majority of the high-quality entries. This reflects their comparatively stronger performance in preliminary generation and evaluation. Further details and representative examples are provided in Appendix A and Appendix G.

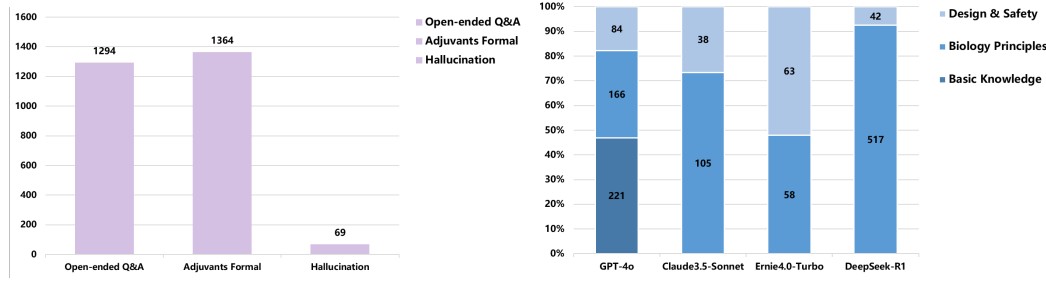

(a) Distribution of the Data Types  (b) Distribution of MLLMs for Generating Q&A Data

Figure 3: Distribution of the Adjuvant Benchmark

#### 3.3.2 OPEN-ENDED Q&A DATA

The open-ended Q&A component is intended to evaluate the extent to which MLLMs capture adjuvant-related knowledge. It draws on both basic and advanced material curated from textbooks and peer-reviewed publications. The advanced category covers two major themes: *biological principles* (e.g., immunological mechanisms of adjuvant action) and *design & safety* (e.g., strategies for developing or modifying adjuvants and approaches for evaluating safety). Table 2 summarizes the distribution of these data, with the advanced subtypes highlighted.

Table 2: Distribution of Open-ended Q&A

| Data Type | Basic Knowledge | Advanced Knowledge | Biology Principles | Design & Safety |
|---|---|---|---|---|
| Count | 221 | 1073 | 846 | 227 |

Table 3: Hallucination Data

| Data Type | Question | Answer | Overlap | Total |
|---|---|---|---|---|
| Hallucination | 27 | 54 | 12 | 69 |

In addition, the benchmark includes multimodal content: 1,135 entries are text-only (87.7%), while 159 involve image-associated inputs (12.3%). This enables evaluation of both purely textual reasoning and multimodal understanding. Illustrative examples are provided in Appendix C.

#### 3.3.3 HALLUCINATION DATA

In this study, the hallucination data follow the same structural format as the open-ended Q&A but differ in that the questions or answers have been reviewed by domain experts and explicitly judged

to be incorrect. Rather than discarding these items, we retain them as a dedicated resource for evaluating the ability of MLLMs to recognize and reject hallucinations in the context of adjuvant and immunology tasks. For clarity, we distinguish two categories: *question hallucinations* and *answer hallucinations*. This dataset provides a controlled setting for analyzing the sources of hallucination errors and offers a reference point for the development of more reliable model evaluation and training strategies.

### 3.3.4 ADJUVANT FORMAL DATA

Formal descriptions are introduced to translate complex biological processes related to adjuvants into structured variables and functional transformations, with the aim of improving both the reasoning capacity and interpretability of MLLMs in this domain. Such formalized pathways also provide a systematic means of representing mechanisms that may otherwise remain implicit or fragmented in the literature.

To construct these descriptions, we worked with the same team of adjuvant experts described in Section 3.2 to design a set of formal variables and functions, thereby establishing expert-defined standards. These standards were organized into templates and incorporated into prompts for GPT-4o, which subsequently generated a total of **1,364** formal entries (Fig. 1c). The data are divided into two balanced categories: *adjuvant design* and *adjuvant activation & immune processes*, each comprising **682** items. Detailed definitions of the variables and functions are provided in Appendix D.

Although this framework has not yet been applied to downstream model training, the released variables and relationships—such as `Form(Struc, Ag)` and `Load(A, B, Surface)`—serve as structured building blocks for future adjuvant-specialized MLLMs. By providing a computable abstraction of design principles and immune response processes, the framework establishes a foundation that can be extended in subsequent research.

## 4 EXPERIMENTS

### 4.1 EXPERIMENTAL SETTINGS

**MLLMs**: The set of evaluated models is listed in Table 4. Models highlighted in blue were also used in the data generation stage and were subsequently reviewed by adjuvant experts.

Table 4: Models evaluated on the adjuvant benchmark. Blue rows indicate models additionally used during data generation.

| Model | #Size | Form | Ver. | Model | #Size | Form | Ver. |
|---|---|---|---|---|---|---|---|
| GPT-4o | N/A | api | latest | Qwen2.5-VL-7B | 7B | open | instruct |
| GPT-4.1 | N/A | api | latest | Qwen2.5-VL-72B | 72B | open | instruct |
| OpenAI-o1 | N/A | api | latest | Qwen3-8B | 8B | open/api | think |
| Claude3.5 | N/A | api | sonnet | Qwen3-32B | 32B | open/api | think |
| Claude3.7 | N/A | api | sonnet | Qwen3-30B-A3B | 30B | open/api | think |
| Gemini1.5-Pro | N/A | api | latest | Qwen3-235B-A22B | 235B | open/api | think |
| Gemini2.0-Pro | N/A | api | flash | Internvl2.5-8B | 8B | open | v2.5 |
| Gemini2.5-Pro | N/A | api | flash | Internvl2.5-78B | 78B | open | v2.5 |
| Ernie3.5 | N/A | api | latest | Internvl3.0-8B | 8B | open | v3.0 |
| Ernie4.0 | N/A | api | turbo | Internvl3.0-72B | 72B | open | v3.0 |
| Doubao1.5-Pro | N/A | api | 250115 | InstructBlip-13B | 13B | open | vicuna |
| DeepSeek-R1 | 671B | open/api | reasoner | Idefics-9B | 9B | open | instruct |
| DeepSeek-V3 | 671B | open/api | chat | Darwin | 7B | open | v1.5 |
| LLaVA1.5-7B | 7B | open | v1.5 | BioGPT | 1.5B | open | Large |
| LLaVA1.5-13B | 13B | open | v1.5 | ScienceOne | 8B | open | Base |

**Inference**: Closed-source models were accessed through their official APIs. For open-source models, inference was performed with the official implementations on $8 \times$ NVIDIA A800 GPUs, following recommended hyperparameter settings. To ensure comparability across models, all were evaluated under identical prompts in a zero-shot setting (see Appendix G.2 for prompt details). To ensure fairness, regardless of whether the model supports multimodal input, we utilized a unified OCR engine to process the images and concatenated the OCR output with the original input text. Furthermore, we assessed the performance of the top 5 multimodal models on the image-related subset, with detailed experimental results provided in Appendix F.

**Evaluation metrics**: To assess knowledge comprehension in the adjuvant domain, we employed a combination of automatic metrics and model-based scoring. Standard measures included Semantic Textual Similarity (STS) and BERTScore. In addition, we introduced an LLM-based rubric, implemented with GPT-4o and DeepSeek-R1, which scored answers along three axes: Similarity Score (SS), Scientific Rationality Score (RS), and Inclusiveness Score (IS), each on a 0–10 scale. This approach provides a scalable and reproducible way to evaluate factual accuracy and conceptual soundness, while reducing dependence on manual annotation. To further probe robustness, we report a Hallucination Rejection Ratio (HRR), which quantifies the ability of models to detect and avoid incorrect content. Detailed metric formulations are given in Appendix E.

## 4.2 EVALUATION OF GENERATION

During the annotation process, experts assigned subjective scores (0–10) to six dimensions: Questioning Ability, Answering Ability, Reasoning Ability, Knowledge Reserve, Chart Analysis, Context Utilization, and Instruction Following. These scores reflect the overall quality and reliability of the generated content.

The results are summarized in Fig. 4a. DeepSeek-R1 obtained comparatively higher scores in both questioning and answering ability, indicating that it can produce relevant prompts and generate responses with coherent reasoning in the adjuvant domain. GPT-4o achieved the highest score in questioning ability and also performed well in instruction following, suggesting that it is effective at generating focused inquiries and adhering to task specifications. In addition, GPT-4o showed broad coverage of domain knowledge, contributing to more comprehensive responses.

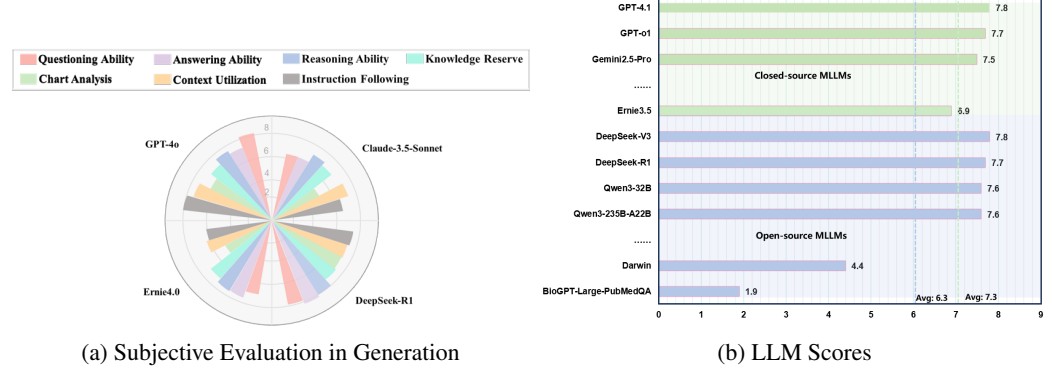

(a) Subjective Evaluation in Generation                    (b) LLM Scores

Figure 4: Comprehensive Evaluation of MLLMs on the Adjuvant Benchmark

By contrast, Ernie4.0 and Claude3.5 received consistently lower scores across several categories, suggesting limitations in handling complex material from adjuvant-related literature.

Overall, the expert assessments highlight GPT-4o and DeepSeek-R1 as the strongest performers within this evaluation setting, particularly in tasks requiring both domain-specific questioning and reasoned answering. These findings provide a basis for selecting suitable base models for future work in adjuvant-focused applications.

## 4.3 EVALUATION OF ADJUVANT OPEN-ENDED Q&A

We evaluated 11 closed-source and 19 open-source MLLMs on the adjuvant open-ended Q&A task. Results are reported in Table 5, with blue highlighting used to indicate models achieving state-of-the-art performance under the given metrics. A comparison between closed- and open-source models is summarized below.

**Closed-source vs. Open-source:** On average, closed-source models achieved higher overall performance, with a mean LLM Score of 7.3 and an STS score of 0.7263, compared to 6.4 and 0.6922 for open-source models. Nevertheless, several open-source models exceeded the closed-source averages. DeepSeek-R1 (LLM Score= 7.7, STS = 0.7415), DeepSeek-V3 (LLM Score = 7.8, STS = 0.7289), Qwen3-235B (LLM Score = 7.6, STS = 0.7331), and Qwen3-32B (LLM Score = 7.6, STS = 0.7259) performed comparably to the strongest closed-source models, particularly in scientific rationality and inclusiveness. However, in terminology consistency, reflected by BERTScore, open-source models

averaged 0.5504, which remains below the closed-source average of 0.5656. This suggests that while optimization strategies enable some open-source models to close the gap, challenges remain in aligning with domain-specific vocabulary. Overall, the observed performance differences appear to relate more to the extent of domain knowledge integration than to the proprietary or open-source nature of the models.

**Inference Models vs. Think Models:** Think-oriented models obtained higher scores in Rationality Score (RS), Inclusiveness Score (IS), and STS compared to inference-style models. This pattern suggests that their explicit reasoning mechanisms—such as multi-step causal decomposition and logical verification—contribute to producing more logically consistent and comprehensive answers. At the same time, the reliance on explicit reasoning chains increases decoding complexity, which may constrain efficiency in resource-limited settings. These observations indicate that combining explicit reasoning strategies with domain knowledge and structured representations could be a promising direction for future model development.

**Closed-source Models:** OpenAI-o1 (LLM Score = 7.7, STS = 0.7495) was the best-performing closed-source model in our setting. By contrast, Ernie4.0 (LLM Score = 6.9) and Doubao1.5-Pro (LLM Score = 7.1) obtained lower scores within this cohort. While performance differences may correlate with factors such as data coverage, training procedures, and model scale, the proprietary nature of these systems prevents attributing causality from our evaluation alone. Closed-source models also pose practical constraints for scientific use, including limited transparency and higher inference costs, which may hinder broad adoption in open research workflows. Future work could explore more transparent and collaborative evaluation practices to facilitate integration into scientific pipelines.

Table 5: Evaluation Result of Adjuvants Open-ended Q&A

| Model Category | STS Score | BERTScore | LLM Score (GPT-4o) | | | LLM Score (DeepSeek-R1) | | | LLM Score Avg |
|---|---|---|---|---|---|---|---|---|---|
| | | | SS | RS | IS | SS | RS | IS | |
| **Closed-source MLLMs** | | | | | | | | | |
| *Inference Models* | | | | | | | | | |
| GPT-4o OpenAI | 0.7261 | 0.5732 | 6.4 | 8.3 | 6.8 | 6.9 | 8.6 | 6.8 | 7.3 |
| GPT-4.1 OpenAI | 0.7178 | 0.5420 | 7.0 | 8.5 | 7.2 | 7.6 | 9.0 | 7.6 | 7.8 |
| Cladue3.5 Claude | 0.7256 | 0.5750 | 6.2 | 8.2 | 6.7 | 6.8 | 8.7 | 6.9 | 7.3 |
| Cladue3.7 Claude | 0.7396 | 0.5650 | 6.5 | 8.2 | 6.8 | 7.0 | 8.7 | 7.1 | 7.4 |
| Gemini1.5-Pro Team et al. (2024) | 0.7235 | 0.5644 | 6.3 | 8.2 | 6.7 | 6.9 | 8.8 | 7.1 | 7.3 |
| Gemini2.0-Pro Google | 0.7118 | 0.5486 | 6.3 | 8.2 | 6.7 | 6.9 | 8.7 | 7.1 | 7.3 |
| Gemini2.5-Pro Google | 0.7316 | 0.5664 | 6.6 | 8.4 | 7.0 | 7.2 | 8.9 | 7.1 | 7.5 |
| Ernie3.5 Baidu | 0.7199 | 0.5554 | 6.1 | 8.0 | 6.5 | 6.4 | 8.2 | 6.2 | 6.9 |
| Ernie4.0 Baidu | 0.7238 | 0.5587 | 6.0 | 8.0 | 6.4 | 6.4 | 8.2 | 6.2 | 6.9 |
| Doubao1.5-Pro Volcengine | 0.7201 | 0.5532 | 6.1 | 8.0 | 6.4 | 6.8 | 8.5 | 6.7 | 7.1 |
| *Think Models* | | | | | | | | | |
| OpenAI-o1 OpenAI | 0.7495 | 0.6195 | 6.9 | 8.5 | 7.1 | 7.3 | 8.9 | 7.2 | 7.7 |
| **Average** | **0.7263** | **0.5656** | **6.4** | **8.2** | **6.8** | **6.9** | **8.7** | **6.9** | **7.3** |
| **Open-source MLLMs** | | | | | | | | | |
| *Inference Models* | | | | | | | | | |
| DeepSeek-V3 Liu et al. (2024a) | 0.7289 | 0.5276 | 6.8 | 8.4 | 7.0 | 7.6 | 9.0 | 7.7 | 7.8 |
| LLaVA1.5-7B Liu et al. (2024b) | 0.7134 | 0.5823 | 5.2 | 7.0 | 5.2 | 5.3 | 6.7 | 4.6 | 5.7 |
| LLaVA1.5-13B Liu et al. (2024b) | 0.7116 | 0.5838 | 5.4 | 7.1 | 5.4 | 5.4 | 6.9 | 4.8 | 5.8 |
| Qwen2.5-VL-7B Bai et al. (2025) | 0.7151 | 0.5602 | 5.8 | 7.8 | 6.1 | 5.9 | 7.7 | 5.7 | 6.5 |
| Qwen2.5-VL-72B Bai et al. (2025) | 0.7217 | 0.5649 | 6.2 | 8.2 | 6.7 | 6.6 | 8.4 | 6.5 | 7.1 |
| Internvl2.5-8B Chen et al. (2024) | 0.7217 | 0.5649 | 5.2 | 7.0 | 5.5 | 5.4 | 6.8 | 5.0 | 5.8 |
| Internvl2.5-78B Chen et al. (2024) | 0.6966 | 0.5606 | 6.0 | 7.8 | 6.3 | 6.2 | 7.9 | 6.0 | 6.7 |
| Internvl3.0-8B Zhu et al. (2025) | 0.6987 | 0.5526 | 5.6 | 7.6 | 6.0 | 6.0 | 7.6 | 5.9 | 6.5 |
| Internvl3.0-78B Zhu et al. (2025) | 0.7173 | 0.5608 | 6.2 | 8.1 | 6.6 | 6.6 | 8.3 | 6.5 | 7.1 |
| InstructBlip-13B Dai et al. (2023) | 0.5960 | 0.5551 | 4.9 | 6.2 | 4.5 | 5.0 | 6.3 | 4.3 | 5.2 |
| Idefics-9B Laurençon et al. (2023) | 0.5662 | 0.4718 | 4.5 | 6.2 | 4.3 | 4.8 | 6.1 | 4.2 | 5.0 |
| *Think Models* | | | | | | | | | |
| DeepSeek-R1 Guo et al. (2025) | 0.7415 | 0.5485 | 6.6 | 8.4 | 7.1 | 7.5 | 9.0 | 7.7 | 7.7 |
| Qwen3-8B Bai et al. (2025) | 0.7275 | 0.5387 | 6.5 | 8.1 | 6.7 | 7.1 | 8.6 | 7.2 | 7.4 |
| Qwen3-32B Bai et al. (2025) | 0.7259 | 0.5371 | 6.6 | 8.1 | 6.9 | 7.3 | 8.8 | 7.6 | 7.6 |
| Qwen3-30B-A3B Bai et al. (2025) | 0.7262 | 0.5411 | 6.5 | 8.3 | 6.9 | 7.2 | 8.8 | 7.3 | 7.5 |
| Qwen3-235B-A22B Bai et al. (2025) | 0.7331 | 0.5497 | 6.5 | 8.4 | 7.0 | 7.3 | 8.9 | 7.6 | 7.6 |
| *Domain-Specific Models* | | | | | | | | | |
| Darwin Xie et al. (2025b) | 0.6376 | 0.6245 | 4.4 | 5.5 | 3.8 | 4.1 | 5.4 | 3.1 | 4.4 |
| BioGPT-Large-PubMedQA Luo et al. (2022) | 0.5468 | 0.4906 | 1.7 | 2.3 | 1.3 | 1.9 | 2.5 | 1.7 | 1.9 |
| ScienceOne Institute of Automation, Chinese Academy of Sciences (2025) | 0.7252 | 0.5408 | 6.9 | 8.5 | 7.0 | 6.8 | 8.5 | 6.9 | 7.4 |
| **Average** | **0.6922** | **0.5504** | **5.7** | **7.4** | **5.8** | **5.9** | **7.5** | **5.8** | **6.4** |

**Open-source Models:** Among the open-source systems, DeepSeek-V3 achieved the highest overall performance. DeepSeek-R1 (LLM Score = 7.7), Qwen3-32B (LLM Score = 7.6), and Qwen3-235B (LLM Score = 7.6) all surpassed the closed-source average (7.3). These models employed explicit reasoning strategies, such as causal decomposition and multi-step verification, which contributed to higher scientific rationality scores and in some cases exceeded those of closed-source models (e.g., Claude3.5). This suggests that open-source models can exhibit strong logical reasoning capabilities, although they often rely on decomposition and iterative reasoning to mitigate limitations in domain-specific knowledge.

DeepSeek-R1 and Qwen3-235B also incorporate Mixture of Experts (MoE) architectures, where dynamic expert routing enables finer-grained knowledge integration. While MoE contributes to improvements in reasoning and task decomposition, terminology consistency remains a challenge:

BERTScore for these models is still lower than that of closed-source models such as GPT-4o. This indicates that MoE and reinforcement learning approaches alone are insufficient, and domain-adaptive pretraining remains necessary for accurate use of specialized terminology.

An additional observation is the non-linear relationship between parameter scale and performance: Qwen3-235B and Qwen3-32B achieved similar LLM Score despite their large difference in size. This pattern highlights diminishing returns from scaling alone and underscores the importance of targeted knowledge injection for domain adaptation. By contrast, models such as InstructBlip-13B and Idefics-9B underperformed across most metrics, reflecting architectural and training-data limitations in earlier generations of multimodal LLMs.

**Domain-Specific Models:** The comparatively lower performance of domain-specific biomedical and materials models indicates limitations in directly transferring such architectures to adjuvant tasks. Their training objectives, often centered on literature summarization or general biomedical QA, are not well aligned with the requirements of adjuvant-focused Q&A, leading to weaker answer quality (see Appendix F.4 for detailed examples).

Notably, ScienceOne, which is fine-tuned on general scientific data based on the Qwen3. While it outperforms its base model on several metrics, it does not achieve a comprehensive lead across all evaluation dimensions. Although integrating general scientific knowledge undoubtedly provides a beneficial foundation for comprehending broader scientific concepts, this reveals a critical limitation: broad knowledge injection alone does not automatically translate to a comprehensive understanding of specialized adjuvant research. To achieve the in-depth comprehension and nuanced reasoning required for adjuvant research, foundational scientific models must undergo further targeted training on domain-specific adjuvant data. These results reinforce the view that progress in adjuvant research requires purpose-built datasets and dedicated models, rather than relying solely on fine-tuning with broader biomedical, materials, or general scientific corpora. Furthermore, the Darwin model obtained the highest BERTScore among all models, which can likely be attributed to its training on synthesized data from the materials and biological domains, enabling it to effectively grasp and utilize highly specialized terminology.

Overall, models from the GPT, DeepSeek, and Qwen3 families demonstrated relatively strong performance across multiple metrics, suggesting that these families already possess the capacity to contribute as auxiliary tools for basic research and as potential foundations for future adjuvant-specialized systems.

## 4.4 EVALUATION OF HALLUCINATION REJECTION

We evaluated the top five models (both closed-source and open-source) based on their LLM Scores for their ability to reject hallucinations. Results are reported in Table 6.

Table 6: Evaluation of Hallucination Rejection Capabilities (Mean ± SD over 10 shuffled evaluation)

| Model Category | Question HRR (%) | Answer HRR (%) | Overall HRR (%) |
|---|---|---|---|
| GPT-4o | **30.74% (± 4.95%)** | **23.33% (± 2.92%)** | **26.23% (± 2.85%)** |
| GPT-4.1 | 22.22% (± 2.47%) | 14.26% (± 1.25%) | 17.10% (± 1.14%) |
| OpenAI-o1 | 24.07% (± 4.70%) | 18.15% (± 2.10%) | 20.58% (± 2.72%) |
| Gemini 2.5 Pro | 18.15% (± 3.68%) | 9.07% (± 1.84%) | 12.32% (± 1.41%) |
| Claude3.7 | 13.33% (± 2.59%) | 22.96% (± 1.79%) | 21.59% (± 1.59%) |
| DeepSeek-V3 | 0.00% (± 0.00%) | 2.69% (± 0.92%) | 2.10% (± 0.72%) |
| DeepSeek-R1 | 22.59% (± 4.08%) | 12.04% (± 3.18%) | 16.23% (± 3.19%) |
| Qwen3-8B | 12.96% (± 4.70%) | 10.37% (± 1.79%) | 11.74% (± 1.74%) |
| Qwen3-32B | 14.81% (± 3.49%) | 8.52% (± 3.17%) | 8.52% (± 3.17%) |
| Qwen3-30B-A3B | 21.11% (± 5.25%) | **17.22% (± 1.96%)** | **18.99% (± 2.41%)** |
| Qwen3-235B-A22B | **23.33% (± 6.06%)** | 16.15% (± 3.53%) | 18.73% (± 3.13%) |

Both closed- and open-source models exhibited limited capability in hallucination rejection. The median HRR for closed-source models was 20.58%, compared to 13.99% for open-source models, which falls below the level generally required for reliable application in practice. For example, DeepSeek-V3 performed strongly on the adjuvant Q&A task (LLM Score = 7.8) but obtained the lowest HRR (2.10%), highlighting the inconsistency between knowledge answering and hallucination rejection.

These findings suggest that current models often rely on surface-level language correlations rather than deeper domain reasoning, which constrains their ability to identify and reject incorrect content. Improving hallucination control in this setting will likely require domain-adaptive fine-tuning combined with structured knowledge representations, in order to enhance logical coherence and scientific reliability. Additional analyses are provided in Appendix F.

## 5    Conclusions and Limitations

This work presents the first benchmark dedicated to adjuvants, combining 1,294 expert-annotated Q&A pairs and 1,364 formal descriptions. Using this resource, we systematically evaluated 11 closed-source and 19 open-source MLLMs across open-ended Q&A, hallucination rejection, and instruction following. Our results highlight comparatively strong performance from the GPT, DeepSeek-R1, and Qwen3.0 families, and we propose a formal framework that abstracts adjuvant design principles and immune mechanisms into structured representations to support future domain-specific models.

While our study provides an initial foundation, further progress will require stratified benchmarks to capture varying task difficulty, domain-adaptive training for expert knowledge integration, and hybrid neuro-symbolic architectures that leverage the proposed formal framework. Beyond technical evaluation, the benchmark and formal abstractions may lower the entry barrier for applying MLLMs in immunology and help systematize reasoning in vaccine adjuvant research. These resources are intended solely for research purposes and should not be used directly in clinical contexts without expert validation.

## 6    Acknowledgement

This work is sponsored by CAS Project for Young Scientists in Basic Research (YSBR-083), Strategic Priority Research Program of Chinese Academy of Sciences (No. XDA0480000, XDA0480200 and XDA0480204), Young Scientists Fund of The State Key Laboratory of Multimodal Artificial Intelligence Systems (No.ES2P100110), 2035 Innovation Mission Project of CASIA (No.E4J10102), National Natural Science Foundation of China (U25A20654), Lingang Laboratory (Grant No. LGL-2616-04), Zhongguancun Academy Project No.02012501.

## Ethics Statement

This paper adheres to the ICLR Code of Ethics. We acknowledge that our research follows ethical guidelines, ensuring compliance with all relevant regulations.

- **Human Subjects:** No human subjects were involved in this research. All data used in the study are publicly available datasets or generated synthetically.
- **Data Privacy and Security:** We have taken care to ensure that all data used complies with relevant data protection laws. Datasets used in this research do not contain any personal, confidential, or sensitive information.
- **Research Integrity:** We confirm that the research presented is original and has not been plagiarized. All sources and data are properly cited and documented.
- **Legal Compliance:** We confirm that the research complies with all applicable laws and ethical standards, including data usage and research practices.

This statement is meant to address potential concerns in accordance with the ICLR Code of Ethics.

## Reproducibility Statement

We ensure that our work is reproducible by providing all necessary resources for others to replicate our experiments. Specifically:

- **Benchmark Data and Experimental Setup:** All datasets used in the experiments, along with detailed data processing steps, are publicly available. The experimental setup is also

provided, including hyperparameters, evaluation metrics, and the benchmarking environment.

- **Source Code:** The code used to process the data and conduct the benchmarking experiments is available in an anonymous repository. This includes all scripts necessary for replicating the experiments as described in the paper.

By providing these resources, we aim to make our results fully reproducible and facilitate further research based on our work. The details required for reproduction can be found in the anonymous repository.

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

APPENDIX OVERVIEW

This appendix provides supplementary material to support reproducibility and clarity. It is organized as follows:

- **Appendix A: Dataset Construction**
    - Additional statistics, distributions, and category breakdown.
- **Appendix B: Data Preprocessing and Expert Annotation**
    - B.1 Details of the preprocessing pipeline.
    - B.2 Details of expert annotation workflow, quality control procedures, and inter-annotator agreement notes.
- **Appendix C: Representative Q&A Examples**
    - Selected samples from the benchmark illustrating different knowledge categories.
- **Appendix D: Formal Framework**
    - D1. Definitions of formal variables.
    - D2. Definitions of formal functions.
    - D3. Definitions of functional transfer relationship.
- **Appendix E: Evaluation Metrics**
    - E.1 Mathematical formulations of Semantic Textual Similarity (STS).
    - E.2 Mathematical formulations of BERTScore.
    - E.3 Definition of LLM Score and consistency verification between the human expert score.
    - E.4 Mathematical formulations of Hallucination Rejection Ratio (HRR).
- **Appendix F: Experimental Analyses**
    - F.1 Extended results on prompt-following.
    - F.2 The State-of-the-art model category comparisons.
    - F.3 Evaluation in visual-related subsets.
    - F.4 Error visualization analysis of domain-specific models.
- **Appendix G: Prompts**
    - G.1 Data generation prompt templates.
    - G.2 Evaluation prompt templates.
- **Appendix H: Word Cloud Visualization**
    - Supplementary visualizations of key concepts and terminology distribution.
- **Appendix I: Usage of Large Language Models.**
    - The usage of Large Language Models in this paper.

## A  DETAILED DATA FOR CHART

### A.1  DISTRIBUTION OF THE ADJUVANT BENCHMARK

Table 7: Distribution of the Data Types

| Category | Open-ended Q&A | Hallucination | Adjuvants Formal |
|---|---|---|---|
| **Count** | 1294 | 69 | 1364 |

### A.2  DISTRIBUTION OF MLLMs FOR GENERATING Q&A DATA

### A.3  SUBJECTIVE EVALUATION OF MLLMs IN GENERATION

Table 8: Distribution of MLLMs for Generating Q&A Data

| Model | DeepSeek-R1 | GPT-4o | Claude3.5-Sonnet | Ernie4.0-Turbo |
|---|---|---|---|---|
| Count | 559 | 471 | 143 | 121 |

Table 9: Subjective Evaluation of MLLMs in Generation

| Ability | GPT-4o | Cladue3.5 | Erine4.0 | DeepSeek-R1 |
|---|---|---|---|---|
| Questioning | **8.0** | 6.4 | 6.9 | 7.9 |
| Answering | 7.0 | 6.5 | 7.4 | **8.1** |
| Reasoning | 7.7 | 7.1 | 7.7 | **8.0** |
| Knowledge Reserve | 7.0 | 6.9 | 7.1 | **7.5** |
| Chart Analysis | 6.2 | 5.3 | 5.0 | **7.7** |
| Context Utilization | 7.2 | 7.1 | 6.1 | **7.8** |
| Instruction Following | **8.0** | 6.5 | 6.0 | 7.5 |

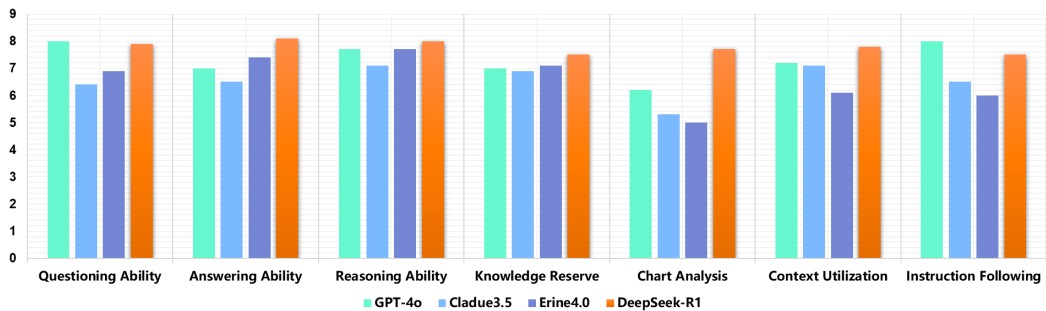

Figure 5: Visualization of Subjective Evaluation in MLLMs Generation

## B    DATA PREPROCESSING AND EXPERT ANNOTATION DETAILS

### B.1    DATA PREPROCESSING

The raw corpus consisted of 739 peer-reviewed research articles and two classic textbooks in immunology and vaccine adjuvants. We employed a document parsing pipeline based on `PyMuPDF + pytesseract` to extract structured text while preserving paragraph hierarchy and separating embedded figures. Both extracted text and figures were then provided as context to multimodal models (e.g., GPT-4o). For unimodal models such as DeepSeek-R1, figures were processed with an OCR engine to obtain textual content, ensuring consistency across evaluations (Fig. 6). All outputs were subsequently reviewed, and only accurate and relevant Q&A pairs were retained in the benchmark.

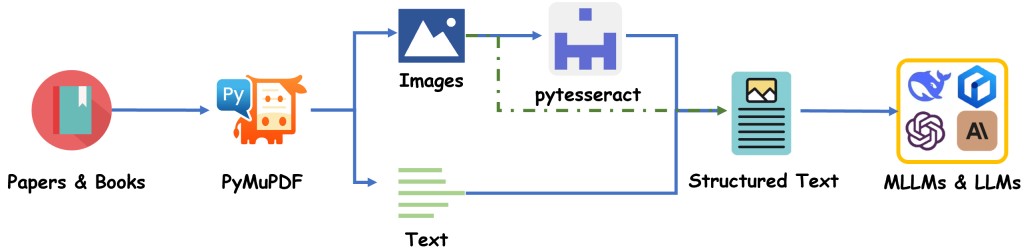

Figure 6: Data preprocessing pipeline.

From the processed corpus, 1,500 candidate Q&A items were randomly sampled for expert review.

### B.2    EXPERT ANNOTATION

#### B.2.1    EXPERT TEAM

Annotation was carried out by 13 specialists spanning complementary areas of vaccine research:

- **Infectious disease vaccines:** 1 senior researcher, 2 PhD students, 3 MSc students.
- **Cancer vaccines:** 1 researcher, 2 PhD students, 2 MSc students.
- **Bacterial vaccines:** 2 MSc students.

All annotators had domain training in immunology or vaccine-related research.

#### B.2.2    EXPERT ANNOTATION WORKFLOW

Annotation followed standardized guidelines co-developed by AI and immunology experts. The workflow was:

1. Define goals: establish the first domain-specific benchmark for adjuvant research.
2. Curate source material: collect high-quality PDFs of papers and textbooks.
3. Determine data requirements: textbooks for foundational knowledge; research articles for advanced content (mechanisms, design principles, safety).
4. Pilot phase: generate trial items with MLLMs; refine through expert feedback.
5. Batch generation: perform large-scale API-based generation once validated.
6. Annotation protocol: assess each item for (i) correctness of the question, (ii) correctness of the answer, (iii) validity of reasoning, (iv) overall quality.  → Incorrect questions ⇒ *question hallucinations*; incorrect answers ⇒ *answer hallucinations*.
7. Standardization: annotators trained to rely exclusively on the source text, avoiding subjective inference.
8. Validation: first-round expert review followed by quality control from AI researchers before later batches.

### B.2.3 CONSISTENCY ASSURANCE

To guarantee reliability and minimize subjectivity, we adopted the following measures:

- **Reference standard:** The original source text was defined as the sole criterion for correctness, preventing reliance on prior knowledge or subjective inference.
- **Unified training:** All annotators underwent standardized training and participated in a trial phase before formal labeling.
- **Joint calibration:** Approximately 30% of the samples were jointly annotated to align interpretations across experts.
- **Independent labeling with discussion:** The remaining 70% of samples were labeled independently, with ongoing discussions to resolve uncertainties.

This multi-step protocol ensured consistent, transparent, and reproducible labeling across the benchmark.

## C  CASES FROM ADJUVANT BENCHMARK

**Question:** What mechanisms contribute to the diversity of the antigen receptor repertoire in lymphocytes?

**Reasoning Process:** The diversity of antigen receptors is generated through somatic recombination, which includes rearrangements of gene segments and the addition of nucleotides.

**Answer:** Somatic recombination, combinatorial diversity, junctional diversity, and somatic hypermutation contribute to the diversity of antigen receptors.

Figure 7: A Case from Basic Knowledge Open-ended Q&A

**Question:** How does the incorporation of MPLA and αGC into the lipid bilayer affect the kinetics and efficacy of the immune response?

**Reasoning Process:** MPLA and αGC are lipophilic adjuvants incorporated into the lipid bilayer of the particles. MPLA serves as a TLR4 agonist, promoting sustained immune responses through activation of dendritic cells and B cells. αGC, a glycolipid, stimulates invariant natural killer T cells through CD1d presentation, leading to rapid antibody production. These adjuvants enhance both the magnitude and duration of the antibody response by engaging different immune pathways.

**Answer:** MPLA enhances sustained immune responses by activating TLR4 pathways, while αGC promotes rapid antibody production through invariant NKT cell activation. Their incorporation into the lipid bilayer results in a more effective and durable immune response by engaging complementary immune pathways.

Figure 8: A Case from Advanced Knowledge (Biological Principles) Open-ended Q&A

**Question:** What structural features of the VLPs are revealed by electron microscopy, and why are these important in Fig?

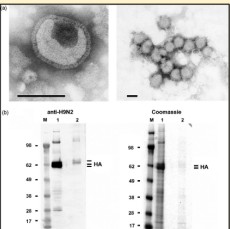

**Reasoning Process:** Electron microscopy images show the size and surface features of VLPs, such as spikes that mimic natural influenza virions, indicating proper assembly and potential for effective antigen presentation.

**Answer:** Electron microscopy reveals VLPs with a diameter of 80–120 nm and surface spikes, indicative of proper assembly and potential to mimic natural virions for effective antigen presentation.

Figure 9: A Case from Advanced Knowledge Vision-related Open-ended Q&A

**Question:** What does Fig reveal about the cellular immune response to MP-adjuvanted vaccines?

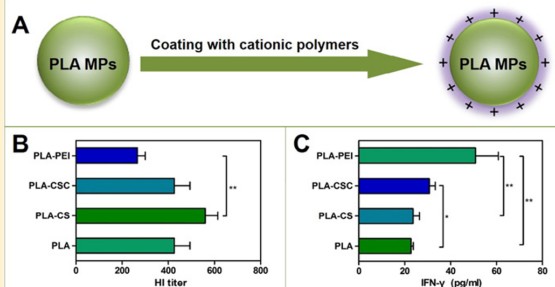

**Reasoning Process:** Understanding cellular immune response data is crucial for evaluating vaccine efficacy.

**Answer:** Fig reveals the levels of IFN-γ produced, indicating the cellular immune response elicited by MP-adjuvanted vaccines with different surface charges, suggesting that more positively charged particles may induce a stronger cellular response.

Figure 10: A Case from Advanced Knowledge Vision-related Open-ended Q&A

**Question:** How do the material properties of PBC micelles (e.g., size, charge, CMC) contribute to their adjuvant function?

**Reasoning Process:** 1. PBC micelles have a 30.4 nm diameter, +5.82 mV zeta potential, and CMC of 0.3 μg/ml. 2. Cationic charge promotes Ag binding and APC uptake. 3. Sub-CMC unimers lack micellar structure, failing to cross-link BCRs.

**Answer:** PBC micelles' nanoscale size (30.4 nm) enables efficient lymphatic drainage and APC uptake. The cationic charge (+5.82 mV) facilitates electrostatic Ag binding and BCR interaction. A low CMC (0.3 μg/ml) ensures micelle stability in physiological conditions, whereas unimers (below CMC) cannot scaffold Ag. These properties collectively enable efficient BCR cross-linking and humoral responses.

Figure 11: A Case from Advanced Knowledge (Adjuvant Design) Open-ended Q&A

**Question:** What are the potential safety concerns associated with PEG–PAEEP, and how are they addressed?

**Reasoning Process:** Safety concerns for PEG–PAEEP include hemolysis and cytotoxicity. The study evaluates these aspects by comparing the hemolytic activity and cytotoxicity of PEG–PAEEP with those of aluminum adjuvants, demonstrating that PEG–PAEEP has significantly lower hemolytic activity and comparable or lower cytotoxicity at effective concentrations.

**Answer:** PEG–PAEEP demonstrates lower hemolytic activity and comparable or lower cytotoxicity than aluminum adjuvants, addressing potential safety concerns effectively.

Figure 12: A Case from Advanced Knowledge (Adjuvant Safety) Open-ended Q&A

**Question:** What safety and usability considerations are addressed for these cpG-loaded particles?

**Reasoning Process:** The use of redox-responsive, biodegradable polypeptide carriers ensures that the particles can be safely degraded after achieving their function. Additionally, the controlled release of cpG minimizes systemic exposure, reducing the risk of adverse effects associated with widespread immune activation.

**Answer:** Safety and usability are enhanced by using biodegradable carriers and controlled release to minimize systemic exposure and potential side effects.

**Reason for Error:** The article does not mention the controlled release of cpG by the particles. Although particulate adjuvants often have the capability for controlled release, this is not the case in this article. This reflects that the general model has a deep understanding of conventional common knowledge, but still falls short in in-depth exploration of the specific article.

Figure 13: A Case from Question Hallucination Generation by GPT-4o

**Question:** How did the experimental design ensure that the observed immune responses were due to the multimerization of antigens on nanoparticles?

**Reasoning Process:** The design involved comparing immune responses from mice immunized with soluble antigens, homotypic nanoparticles, and mosaic nanoparticles. The inclusion of different control groups allowed the researchers to isolate the effects of antigen multimerization on immune response.

**Answer:** The experimental design included controls such as soluble antigen immunization, which did not elicit cross-reactive responses, thereby demonstrating that multimerization on nanoparticles was responsible for the enhanced immunogenicity observed.

**Reason for Error:** The immune response of soluble antigens and nanoparticle antigens needs to be evaluated by the antibody titer (this is the most important) and cross-reaction.

Figure 14: A Case from Answer Hallucination Generation by GPT-4o

**Question:** How does the design of CpG-loaded particles enhance T cell responses and persistent antibodies?

**Reasoning Process:** The particles are designed to facilitate efficient delivery and prolonged release of CpG to pDCs, which are crucial for initiating strong adaptive immune responses. By activating pDCs, the particles enhance antigen presentation and costimulatory signals, promoting robust T cell activation and aiding the development of long-lasting antibody responses.

**Answer:** Design enhances T cell responses and antibodies by ensuring effective CpG delivery and sustained pDC activation, promoting strong adaptive immunity.

**Reason for Error:** First, the article does not include indicators related to T cells and antibodies, so the question itself goes beyond the scope of the article. While the reasoning might be correct in general, there is not much content in this article to support it, as it only covers enhanced delivery and pDC activation. The answer is even more erroneous, as it mistranslates T cells and antibodies and reverses their order with pDC activation, while also focusing entirely on aspects that were not mentioned in the article.

Figure 15: A Case from Q&A Hallucination Generation by GPT-4o

**Question:** What further research would be needed to develop these RV-VLPs into a viable rabies vaccine candidate?

**Reasoning Process:** While the article demonstrates promising initial results, several additional steps would be needed to develop this into a viable vaccine candidate. These would likely include optimizing production and purification processes, conducting more extensive immunogenicity studies, evaluating protective efficacy against rabies virus challenge, assessing long-term immunity, and conducting safety studies.

**Answer:** Further research needed would likelyinclude:1.optimizing RV-VLP production and purification processes.

Figure 16: A Case from the Invalid Data Generation by Claude3.5: Incomplete Answer

# D  ADJUVANT FORMAL DATA

## D.1  DEFINITION OF ADJUVANT FORMAL VARIABLES AND FUNCTIONS

### D.1.1  FORMAL VARIABLES

**Vaccines** ($V$): The vaccine, if there are multiple new vaccines, they can be defined as $V = \{Vac_1, Vac_2, \ldots, Vac_n\}$.

**Experimental Group** ($EG$): Defined as $Vac_e = EG$.

**Control Group** ($CG$): Defined as $Vac_e = CG$.

**Original Viral Surface Antigen** ($Ag$): The original viral surface antigen defined as $Ag$. If there are multiple antigens, they can be enumerated as $Ag_1, Ag_2, \ldots$.

**Structural Configurations** ($Struc$): If antigens possess specific structural configurations, such as particles or dimers, these structures are described and defined as $Struc$. Multiple structures may be defined as $Struc_1, Struc_2, \ldots$.

**Antigens in New Vaccines** ($Ag$): In new vaccines, the antigens employed are similarly defined as $Ag$.

**Adjuvant Forms** ($Struc$): The forms of adjuvants include small molecules, particles, gels, inorganic salts, vesicles, and others. If the literature explicitly specifies the structural forms of adjuvants (e.g., nanoparticles, microparticles, etc.), these adjuvants should be formally incorporated and defined as $Adj_1$, $Adj_2$, and so forth.

**Movement Variables Related to Vaccine Delivery**: Examples include residence (*Stay at*), drainage (*Drain to*), and targeting (*Target to*), as illustrated below:

*Stay at the injection site / lung / gut / . . .* :

$$\text{Stay at Injection}_{\text{site}}/\text{Lung}/\text{Gut}/\ldots$$

*Drain to lymph nodes / spleen / bone marrow / . . .* :

$$\text{Drain to Lymph}_{\text{node}}/\text{Spleen}/\text{Bone}_{\text{marrow}}/\ldots$$

*Targeted delivery to lymph nodes / spleen / bone marrow / dendritic cells / T cells* :

$$\text{Target to Lymph}_{\text{node}}/\text{Spleen}/\text{Bone}_{\text{marrow}}/\text{DCs}/\text{T}_{\text{cells}}/\ldots$$

**Definitions of Innate Immune Cells**:

*Conventional Dendritic Cells 1 (cDC1)*: $\text{Cell}_1 = \text{cDC1 in injection site}$

*Monocyte-derived Macrophages (MoM)*: $\text{Cell}_2 = \text{MoM in peripheral blood}$

*Tissue-resident Macrophages (TRM)*: $\text{Cell}_3 = \text{TRM in spleen}$

*Neutrophils*: $\text{Cell}_4 = \text{Neutrophils in lymph nodes}$

*Plasmacytoid Dendritic Cells (pDC)*: $\text{Cell}_5 = \text{pDC in peripheral blood}$

*Maturation-induced Macrophages*: $\text{Cell}_6 = \text{Mature Macrophages in tissues}$

**Actions of Innate Immune Cells**:

*Recruitment and Activation of DC*: Recruit / Activate of DC

*Uptake of Antigen / Adjuvant / Other by DC*: Antigen / Adjuvant / ... Uptake of DC

*Secretion of Cytokine / Chemokine / Other by DC*: Cytokine / Chemokine / ... Secret of DC

*Antigen Presentation by DC*: Antigen presentation of DC

*Migration of DC*: Migrate of DC

*Phagocytosis by DC*: Phagocytosis of DC

*Costimulation of T cells by DC*: Costimulation of T cells by DC

*Cytotoxic activity of DC*: Cytotoxic activity of DC

**Definitions of Various T Cells**:

*Follicular Helper T Cells (Tfh)*: $Cell_1$ = Tfh in lymph node

*CD4+ T Cells*: $Cell_2$ = CD4 T cell in peripheral blood

*CD8+ T Cells*: $Cell_3$ = CD8 T cell in spleen

*Th1 Cells*: $Cell_4$ = Th1 in lymph node

*Th2 Cells*: $Cell_5$ = Th2 in spleen

*Memory T Cells*: $Cell_6$ = Memory T cell in peripheral blood

**Actions of T Cells**:

*Proliferate*: Movement = Proliferate of Cell

*Activate*: Movement = Activate of Cell

*Differentiate*: Movement = Differentiate of Cell

*Secret*: Movement = Secrete of Cell

*Migrate*: Movement = Migrate of Cell

*Mutate*: Movement = Mutate of Cell

**Definitions of B Cells**:

*Germinal Center B Cells (GCB)*: $Cell_1$ = GCB in lymph node and spleen

*Plasma Blasts*: $Cell_2$ = Plasma Blast in peripheral blood and lymph node

*Plasma Cells*: $Cell_3$ = Plasma Cell in bone marrow

*Memory B Cells*: $Cell_4$ = Memory B cell in bone marrow and lymph node

*Long-Lived Plasma Cells*: $Cell_5$ = Long-Lived Plasma Cell in bone marrow

**Actions of B Cells**:

*Proliferate*: Movement = Proliferate of Cell

*Activate*: Movement = Activate of Cell

*Differentiate*: Movement = Differentiate of Cell

*Migrate*: Movement = Migrate of Cell

*Mutate*: Movement = Mutate of Cell

**Definition of Antibodies:**

Antibodies have three defining criteria: the source of the body fluid (such as serum or bronchoalveolar lavage fluid), the target antigen (defined according to the structure of the antigen established in the first step), and the type (such as IgG, IgA, or simply Antibody, which must refer to the terminology used in the literature).

Each type of antibody must be defined structurally, following the format:

$$Ab = \text{``Type''} \text{ to ``Antigen'' in ``Body Fluid Type''}$$

Example: The antibody IgG to antigen Ag in serum:

$$Ab = \text{IgG to Ag in Serum}$$

**Actions of Antibodies**:

*Secrete*: - Movement refers to the increase in antibody titers, which can be interpreted as an increase in antibody secretion.

$$\text{Movement = Secrete of Ab}$$

*Affinity*: - Movement refers to the enhanced binding of antibodies to viruses, interpreted as an increase in affinity.

$$\text{Movement = Affinity of Ab}$$

*Cross-reactivity*: - Movement refers to the enhanced binding of antibodies to antigens from other variants, interpreted as an increase in cross-reactivity.

$$\text{Movement = Cross-reactivity of Ab}$$

*Neutralization*: - Movement refers to the ability of antibodies to block pathogenic activity through target engagement.

$$\text{Movement = Neutralization by Ab}$$

**Definitions of Other Immune Reactions:**

In the text, there may be some summary-type descriptions of immune responses, such as GC responses and T cell responses. Sometimes, it is not appropriate to define them as the above movement variable, instead, these immune responses can be directly defined as movement variables.

**Example:** The GC response in the lymph node:

$$\text{Movement = GC responses in lymph node}$$

### D.1.2 FORMAL FUNCTION

**Representation of Composition ($Form$):** If an antigen $Ag_1$ forms a structure $Struc_1$, it can be expressed as:

$$Form(Struc_1, Ag_1)$$

The function $Form(A, B/C/D/\ldots)$ represents the composition and refers to the assembly of substances or antigens $B/C/D/\ldots$ into $A$ (adjuvants, structures, etc.).

**Loading**: The function $Load(A, B, \text{Inside/Surface/}\ldots)$ refers to loading $A$ into the inside or on the surface of $B$.

$$Load(A, B, \text{Inside}) \quad \text{or} \quad Load(A, B, \text{Surface})$$

**Mixing**: The function $Mix(A, B)$ refers to simply blending $A$ and $B$ together.

$$Mix(A, B)$$

**Chemical Coupling**: The function $Link(A, B)$ refers to chemically connecting $A$ to $B$ via chemical bonds, protein linkage systems, or linkers.

$$Link(A, B)$$

**Custom Combination Method**: If new combination methods are specified in the literature, they should be defined appropriately, for example, a function $Combine(A, B)$ may represent a new method of combination.

$$Combine(A, B)$$

### D.1.3 FUNCTIONAL TRANSFER RELATIONSHIP

**Comparative Relationships Between Experimental Group and Control Group:**

**Example 1:** Experimental group EG enhances the action (Movement variable) compared to the control group CG.

$$\text{EG Enhance to CG at }\ldots/\ldots$$

**Example 2:** Experimental group EG reduces the action (Movement variable) compared to the control group CG.

$$\text{EG Reduce to CG at }\ldots/\ldots$$

**Persistent Comparison:** For continuous comparative relationships, only when the original text explicitly contains words such as *prolong*, *extend*, *persistent*, *sustained*, or *durable*, can the persistent comparative relationship be established.

**Example 3:** Experimental group EG prolongs the action (Movement variable) compared to the control group CG.

<p align="center">EG Prolong to CG at . . ./. . .</p>

**Transfer Relationships:**

*Causal relationship:* For causal relationships that are explicitly stated or indicated in the article, use the symbol **>>** to represent them.

*Comparative relationships and the conjunction:* Use the symbol **&** to represent comparative relationships and the conjunction of immune pathways.

*Further extend causal relationships:* After establishing a clear causal inference, it is possible to further extend this causal chain based on existing background knowledge. However, only the actions and indicators explicitly stated in the article may be used, and no new actions can be introduced. Only the existing causal relationships between actions may be supplemented, represented by the symbol **-->**.

## D.2 THE CASES OF ADJUVANT FORMAL DATA

---

**Adjuvant Formal Language:**

```
Ag_1 = Trp2 peptide from melanoma
Adj_1 = PSA micelle
Struc_1 = Cationic micelle
Ag_2 = Form(Struc_1, Ag_1)
Adj_1 = Form(Struc_1, polyethylenimine (PEI-2k)/stearic acid)
Vac1 = Load(Ag_2, Adj_1, Inside) = EG
Vac2 = Mix(Ag_1, Adj_1) = CG1
Vac3 = Ag_1 = CG2
```

**Natural language Text:** The study introduces a novel vaccine formulation where PSA micelles are utilized to deliver the Trp2 melanoma antigen peptide. These micelles demonstrate an average size of 28.7 ± 8.2 nm with a near-perfect encapsulation efficiency of the antigen. The PSA micelles are prepared by conjugating branched PEI-2k with stearic acid, forming an amphiphilic structure. The PSA micelles enhance antigen-specific CTL responses and show a preferential accumulation in draining lymph nodes, thereby minimizing systemic toxicity. This vaccine is compared against controls such as free Trp2 and a mixture of Trp2 with empty PSA micelles to assess its efficacy in enhancing immune responses and inhibiting tumor growth in a murine melanoma model.

---

Figure 17: A Case from Adjuvant Formal about Adjuvant Design

**Adjuvant Formal Language:**

*Adjuvant Definition:* Adj = PEG–PAEEP = EG

*Control Group Definition:* CG = Aluminum adjuvants

*Immune Component Definitions:*

Movement_1 = HBsAg-specific IgG titers

Cell_1 = IgG-producing B cells in bone marrow

Movement_2 = Differentiate of Cell_1

*Combining Comparison Relationships and Immune Components:*

EG Enhance to CG at Movement_1

EG Enhance to CG at Movement_2

*Causal Relationship Inference:*

EG Enhance to CG at Movement_1 >> EG Enhance to CG at Movement_2

*Comparative Relationships Extraction to Simplify Causal Chains:*

EG to CG

Enhance(Movement_1) >> Enhance(Movement_2)

**Natural language Text:** The PEG–PAEEP copolymer enhances the immune responses of the HBsAg-VLP vaccine compared to aluminum adjuvants, inducing significantly higher HBsAg-specific IgG titers in mice after the second immunization.

Figure 18: A Case from Adjuvant Formal about Adjuvant Activation and Immune Process

# E  EVALUATION METRICS

Traditional n-gram–based metrics, which rely on character or token overlap, are not well suited to open-ended question answering. They emphasize surface similarity and often miss deeper semantic alignment. We therefore adopt **Semantic Textual Similarity (STS)** and **BERTScore**, which operate at the semantic level rather than raw overlap. In addition, we report an **LLM-based score** that complements embedding metrics by explicitly rating answers along predefined rubrics (similarity, rationality, inclusiveness). Together, these metrics provide complementary perspectives and allow for more reproducible and transparent evaluation.

To assess hallucination rejection, we leverage hallucination data from the adjuvant benchmark. Models are prompted with expert-annotated incorrect Q&A and asked to judge whether a sample is invalid; the resulting *hallucination rejection ratio* (HRR) measures the proportion of correctly rejected items.

## E.1  SEMANTIC TEXTUAL SIMILARITY (STS)

STS evaluates the semantic proximity between two texts via cosine similarity of sentence embeddings. **We compute embeddings with the `SentenceTransformer` Python module (model: `all-mpnet-base-v2`)** Reimers & Gurevych (2019).

Given texts $T_1$ and $T_2$ with embeddings $\mathbf{E}(T_1)$ and $\mathbf{E}(T_2)$, the score is

$$\text{STS}(T_1, T_2) = \frac{\mathbf{E}(T_1) \cdot \mathbf{E}(T_2)}{\|\mathbf{E}(T_1)\| \, \|\mathbf{E}(T_2)\|}. \tag{1}$$

The value ranges from $-1$ to $1$, with larger values indicating stronger semantic alignment.

## E.2  BERTSCORE

BERTScore computes token-level semantic similarity using contextual embeddings. For a generated text $G$ and a reference $C$, we form a similarity matrix

$$S_{i,j} = \frac{\mathbf{E}(G_i) \cdot \mathbf{E}(C_j)}{\|\mathbf{E}(G_i)\| \, \|\mathbf{E}(C_j)\|}, \tag{2}$$

take $P(i) = \max_j S_{i,j}$ as the best match for token $G_i$, and average:

$$\text{BERTScore} = \frac{1}{|G|} \sum_{i=1}^{|G|} P(i). \tag{3}$$

Scores are normalized to $[0, 1]$ (higher is better). Compared with STS (sentence-level semantics), BERTScore emphasizes token-level precision. **For our BERTScore calculations, we use the `scibert_scivocab_uncased` model** Beltagy et al. (2019) **with its corresponding tokenizer**.

## E.3  LLM SCORE

To complement embedding-based metrics, we employ an **LLM-based evaluation** with GPT-4o and DeepSeek-R1. Each candidate answer is assessed along three dimensions:

- **Similarity Score (SS):** factual alignment with the expert reference answer.
- **Rationality Score (RS):** scientific soundness and logical coherence of the reasoning process.
- **Inclusiveness Score (IS):** coverage of essential points and completeness in addressing the question.

Each dimension is scored on a $[0, 10]$ scale, with the following interpretation in the context of adjuvant knowledge:

- **0–3 (poor):** major factual errors or incoherent reasoning, reflecting a lack of basic understanding of adjuvant concepts.

- **4–6 (adequate):** partially correct and logically consistent answers, but with noticeable gaps or oversimplifications in immunological mechanisms or design principles.
- **7–10 (strong):** scientifically consistent, well-reasoned, and comprehensive answers that demonstrate a solid grasp of adjuvant biology and related immunological processes.

These three dimensions are deliberately chosen for the adjuvant domain: factual alignment (SS) captures accuracy of immunological details, rationality (RS) reflects whether the explanation is mechanistically plausible, and inclusiveness (IS) ensures that answers go beyond isolated facts to integrate the multifaceted nature of adjuvant design and immune modulation.

*Why LLM Score is valid:* Using large language models as evaluators ("LLM-as-a-judge") has become a widely adopted practice in open-ended evaluation. Prior studies in general NLP benchmarks demonstrate that LLM-based judgements correlate strongly with human preferences when prompts and rubrics are standardized, and also analyze typical biases such as verbosity or self-preference Zheng et al. (2023); Dubois et al. (2024); Kim et al. (2024); Panickssery et al. (2024); Xie et al. (2025a).

Beyond general-purpose tasks, recent work shows that LLM-based evaluation is also effective in scientific and biomedical domains. For instance, scientific question answering and biomedical information extraction have employed rubric-guided LLM judges to approximate expert assessment D'Souza et al. (2025); Laskar et al. (2025). In the medical domain, Wang et al. (2025a) integrated LLM evaluators into a formal framework for clinical ambient scribing, published in a Nature journal. Similarly, mathematical reasoning tasks have used LLM judges to assess solution validity under verifiable criteria Stephan et al. (2024). These precedents confirm that LLM-as-a-judge is not only scalable but also increasingly recognized across scientific subfields.

In our setup, we (i) fix prompts and decoding parameters to minimize variance, (ii) average scores from two high-performing evaluators (GPT-4o and DeepSeek-R1) to reduce single-model bias, and (iii) complement LLM-based scores with embedding metrics (STS/BERTScore) for transparency and reproducibility. This multi-perspective design yields a reliable proxy for expert assessment while keeping annotation costs tractable, aligning with best practices reported across both general and scientific domains.

### E.3.1 EXPERIMENT OF CONSISTENCY VERIFICATION BETWEEN LLM SCORE AND HUMAN EXPERT SCORE

To verify the effectiveness of using LLMs as judges in the field of adjuvants, we conducted a consistency test between LLM scores and human expert scores. Specifically, we randomly selected 100 responses from all models, with 50 evaluated by GPT-4o and the other 50 evaluated by DeepSeek-R1. These 100 samples were then submitted to human experts for evaluation, using criteria that were completely consistent with those used for the LLM scores. The results are presented in Table 10.

Table 10: Consistency Verification Between LLM Scores and Human Expert Scores

| Scoring Model | Rating Dimension | Pearson Correlation | Spearman Correlation | Kendall Correlation |
|---|---|---|---|---|
| **GPT-4o** | Similarity | 0.8412 | 0.8469 | 0.7752 |
| | Rationality | 0.8006 | 0.6342 | 0.5793 |
| | Inclusiveness | 0.8135 | 0.7487 | 0.6749 |
| | **Avg** | **0.8407** | **0.8044** | **0.7084** |
| **DeepSeek-R1** | Similarity | 0.9145 | 0.9236 | 0.8185 |
| | Rationality | 0.8767 | 0.8456 | 0.7732 |
| | Inclusiveness | 0.8636 | 0.8749 | 0.7613 |
| | **Avg** | **0.9125** | **0.9247** | **0.8019** |
| **All** | Similarity | 0.8854 | 0.8920 | 0.8019 |
| | Rationality | 0.8443 | 0.7986 | 0.7239 |
| | Inclusiveness | 0.8461 | 0.8416 | 0.7413 |
| | **Avg** | **0.8803** | **0.8871** | **0.7739** |

The experiment and visualization (as shown in Fig 19) demonstrate high reliability in using these models as scorers. Both GPT-4o and DeepSeek show strong linear correlations with expert scores, particularly in the Similarity dimension, where GPT-4o achieves a Pearson correlation of 0.8412 and

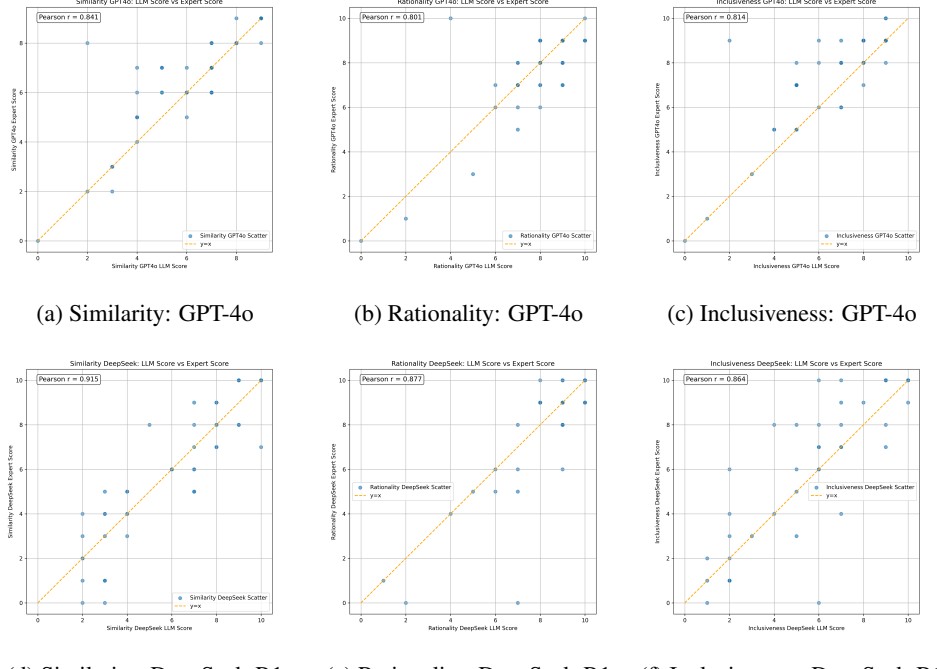

(a) Similarity: GPT-4o     (b) Rationality: GPT-4o     (c) Inclusiveness: GPT-4o

(d) Similarity: DeepSeek-R1    (e) Rationality: DeepSeek-R1    (f) Inclusiveness: DeepSeek-R1

Figure 19: LLM Score vs. Expert Score

DeepSeek reaches 0.9145. Although GPT-4o's correlations are slightly lower, it remains consistently strong across all dimensions, indicating that both models align well with expert evaluations and are effective scoring agents.

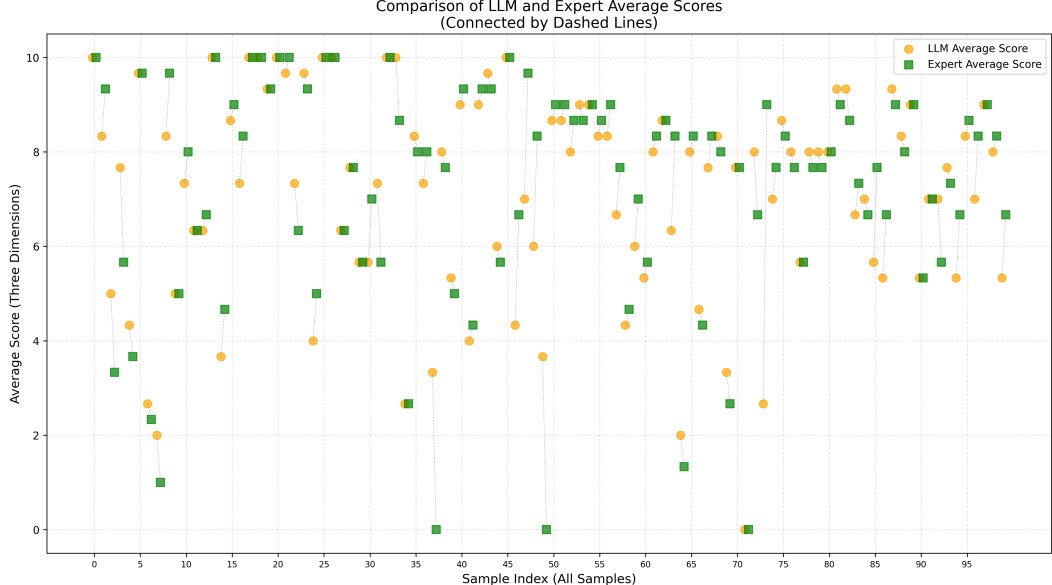

Figure 20: Comparison of LLM and Expert Average Scores

Further analysis using Spearman correlation reveals that DeepSeek outperforms GPT-4o, especially in the Similarity (0.9236) and Rationality (0.8456) dimensions. Despite GPT-4o's lower performance in Rationality (0.6342), it still maintains a reasonable level of ranking consistency, reinforcing its validity as a scorer. The Kendall correlation results mirror these findings, with DeepSeek consistently showing higher scores across all dimensions, particularly in Similarity (0.8185) and Rationality

(0.7732). Although GPT-4o's performance is lower in comparison, it remains within an acceptable range, particularly in Similarity (0.7752), which indicates a strong consistency in ranking with expert scores.

Overall, both GPT-4o and DeepSeek show strong alignment with expert scores, as demonstrated in Fig 20. The average scores across the three dimensions for both the LLM and expert evaluations largely overlap, confirming their reliability as scorers for this task. While DeepSeek slightly outperforms GPT-4o, particularly in Pearson and Spearman correlations, both models exhibit substantial effectiveness and are well-suited for scoring tasks in this domain.

### E.3.2    THE PROMPT OF LLM SCORE

---

**The Prompt of LLM Score Evaluation**

You are an expert in immunology and adjuvant design, with great achievements in immunology and adjuvant design. With the aim of simplifying the thinking process, please score the model's answers and labels (out of 10 points) based on the similarity between the answers and labels, the reasonableness of the answers, and whether the answers incorporate the meaning of the labels. Finally, please provide the results in the following format:
**Similarity Score:** x
**Rationality Score:** x
**Inclusiveness Score:** x

---

### E.4    HALLUCINATION REJECTION RATIO (HRR)

The hallucination rejection ratio (HRR) quantifies a model's ability to detect and resist hallucinated content:

$$\text{HRR} = \frac{\text{Number of correctly rejected hallucinated samples}}{\text{Total hallucinated samples}} \times 100\%.$$

Here, the numerator counts expert-labeled hallucinated Q&A pairs that the model successfully flags as invalid. A higher HRR indicates stronger robustness, which is particularly important in safety-sensitive biomedical applications.

**Prompt design.** The evaluation prompt explicitly defines the label mapping: *"Yes" means the Q&A is scientifically valid, while "No" means it contains hallucination (i.e., invalid claims)*. Additional criteria are listed (factual errors, logical inconsistencies, causal reversal, pseudo-scientific claims, domain irrelevance, or incomplete answers). In practice, the model is instructed to return "No" whenever any of these problems occur, and "Yes" only when the Q&A is entirely correct and domain-relevant. The requirement to "strictly return Yes or No" ensures output consistency and facilitates automatic scoring; it does not prevent verification, since the ground truth hallucination labels were already provided by domain experts during dataset construction.

**Sample size.** The current hallucination dataset consists of 69 high-quality, expert-annotated examples. While relatively small, it represents the first curated hallucination resource in the adjuvant domain. To mitigate sample-size concerns, we report mean HRR with standard deviations over ten repeated trials. This provides an uncertainty estimate and ensures that performance differences reflect systematic model behavior rather than random variation.

Together, this setup ensures that HRR measures what it is intended to capture: whether models can resist generating or endorsing scientifically invalid claims in immunology and adjuvant science.

### E.4.1 THE PROMPT OF HALLUCINATION EVALUATION

---

**The Prompt of Hallucination Evaluation**

You are an expert in immunology and adjuvant design, with great achievements in the immunology and adjuvant fields. Determine if the provided **Question** or its corresponding **Answer** contains scientifically invalid claims in the fields of **Biology**, **Immunology**, or **Adjuvant Science**. Return **only "Yes" (valid)** or **"No" (invalid)** without explanations.

**Input Format:**
- **Question: [Insert question text]**
- **Answer: [Insert answer text]**

**Evaluation Criteria**
Return **"No"** if **ANY of the following apply (including but not limited to)**:
**1.Factual Errors**
- **Contradicts established scientific consensus.**
**2.Logical Inconsistencies**
- **Self-contradictory statements.**
**3.Causal Reversal**
- **Inverts cause-effect relationships.**
**4.Pseudo-Scientific Claims**
- **Unproven theories.**
**5.Domain Irrelevance**
- **Topics outside biology/immunology/adjuvant science.**
**6.Incomplete Answers - Missing critical steps/mechanisms.**

Return "Yes" ONLY **if the Q&A pair is scientifically accurate, logically consistent, and domain-relevant**.

**Response Requirement:**
- **Strictly return "Yes" or "No" in a single line.**
- **No markdown, formatting, or additional text.**

---

# F  EXPERIMENT ANALYSIS

## F.1  EVALUATION OF PROMPT-FOLLOWING

We assessed the prompt-following ability of 11 closed-source and 18 open-source MLLMs under identical hyperparameter settings. Results are summarized in Table 11.

**Closed-source vs. open-source.** Closed-source models consistently outperform open-source counterparts, showing higher average STS and BERTScore when prompts are introduced. This indicates stronger instruction parsing and more reliable adherence to task requirements.

**Performance shifts in open-source models.** Interestingly, while prompts generally improved semantic alignment (higher STS/BERTScore), several open-source models exhibited a *drop in overall LLM Score*. Manual inspection suggests that prompts sometimes pushed these models toward verbose or rigidly formatted answers, inflating token-level similarity but reducing factual soundness and completeness. This mismatch highlights a trade-off between surface-level semantic alignment and deeper reasoning accuracy in open-source systems.

**Model-specific variability.** Domain-specific systems such as BioGPT-Large perform particularly poorly without explicit guiding phrases, in some cases even degrading compared to their no-prompt baseline. For example, BioGPT-Large often required a deterministic prefix (e.g., `"The answer is:"`) to produce stable and interpretable outputs. Without such hints, its responses tended to diverge from the expected format, which explains its negative improvement in Table 11 and 5.

**inference Model vs. Think Model.** "Think" models with explicit reasoning mechanisms show relatively stable performance when prompts are added, compared to inference-only models. This stability likely stems from their multi-step reasoning pathways, which already encourage adherence to task constraints. By contrast, inference models exhibit more variability, suggesting that they are more sensitive to prompt phrasing. Notably, gains in STS and BERTScore for both families are partly explained by prompts eliciting more domain-specific terminology, which boosts surface alignment metrics.

**Summary.** These findings highlight prompt-following as a key dimension of model robustness. However, improvements in surface-level metrics do not always translate to better overall judgment (LLM Score), particularly for open-source systems. This underscores the need for evaluation frameworks that disentangle genuine reasoning gains from superficial prompt-induced artifacts.

Table 11: Evaluation of Adjuvants in Open-ended Q&A (Without prompt)

| Model Category | STS Score | BERTScore | LLM Score (GPT-4o) | | | LLM Score (DeepSeek-R1) | | | LLM Score Avg |
|---|---|---|---|---|---|---|---|---|---|
| | | | SS | RS | IS | SS | RS | IS | |
| **Closed-source MLLMs** | | | | | | | | | |
| *Inference Models* | | | | | | | | | |
| GPT-4o | 0.7190 | 0.5693 | 6.3 | 8.2 | 6.6 | 6.7 | 8.5 | 6.5 | 7.1 |
| GPT-4.1 | 0.7150 | 0.5219 | 6.6 | 8.1 | 6.9 | 7.4 | 8.9 | 7.4 | 7.7 |
| Cladue3.5 | 0.7153 | 0.5598 | 5.8 | 7.8 | 6.2 | 6.4 | 8.6 | 6.6 | 6.9 |
| Cladue3.7 | 0.7323 | 0.5596 | 6.3 | 8.0 | 6.5 | 6.8 | 8.6 | 6.8 | 7.2 |
| Gemini1.5-Pro | 0.7123 | 0.5596 | 6.2 | 8.1 | 6.4 | 6.7 | 8.6 | 6.7 | 7.1 |
| Gemini2.0-Pro | 0.6927 | 0.5362 | 6.0 | 7.9 | 6.3 | 6.6 | 8.5 | 6.8 | 7.0 |
| Gemini2.5-Pro | 0.6927 | 0.5362 | 6.5 | 8.2 | 6.8 | 7.1 | 8.7 | 7.0 | 7.4 |
| Ernie3.5 | 0.7121 | 0.5502 | 5.8 | 7.8 | 6.2 | 6.2 | 8.0 | 6.0 | 6.7 |
| Ernie4.0 | 0.7337 | 0.6027 | 6.1 | 7.7 | 6.1 | 6.1 | 7.9 | 5.6 | 6.6 |
| Doubao1.5-Pro | 0.7093 | 0.5490 | 6.0 | 7.8 | 6.2 | 6.6 | 8.3 | 6.6 | 6.9 |
| *Think Models* | | | | | | | | | |
| OpenAI-o1 | 0.7310 | 0.5818 | 6.9 | 8.5 | 7.1 | 7.3 | 8.9 | 7.3 | 7.7 |
| **Average** | **0.7150** | **0.5569** | **6.2** | **8.0** | **6.5** | **6.7** | **8.5** | **6.7** | **7.1** |
| **Open-source MLLMs** | | | | | | | | | |
| *Inference Models* | | | | | | | | | |
| DeepSeek-V3 | 0.7255 | 0.5254 | 6.5 | 8.2 | 6.8 | 7.4 | 8.8 | 7.5 | 7.5 |
| LLaVA1.5-7B | 0.7115 | 0.5763 | 5.1 | 6.5 | 4.8 | 5.0 | 6.2 | 4.3 | 5.3 |
| LLaVA1.5-13B | 0.7140 | 0.5888 | 5.3 | 7.0 | 5.2 | 5.4 | 6.8 | 4.7 | 5.7 |
| Qwen2.5-VL-7B | 0.7008 | 0.5513 | 5.5 | 7.6 | 5.8 | 5.7 | 7.5 | 5.4 | 6.3 |
| Qwen2.5-VL-72B | 0.7054 | 0.5723 | 5.7 | 7.5 | 5.9 | 5.9 | 7.8 | 5.7 | 6.4 |
| Internvl2.5-8B | 0.6824 | 0.5606 | 5.0 | 6.7 | 5.0 | 5.1 | 6.4 | 4.6 | 5.5 |
| Internvl2.5-78B | 0.7156 | 0.5675 | 5.8 | 7.6 | 5.9 | 6.0 | 7.6 | 5.7 | 6.4 |
| Internvl3.0-8B | 0.7111 | 0.5657 | 5.4 | 7.3 | 5.5 | 5.5 | 7.1 | 5.2 | 6.0 |
| Internvl3.0-78B | 0.7198 | 0.5679 | 6.0 | 7.8 | 6.2 | 6.2 | 8.0 | 6.0 | 6.7 |
| InstructBlip-13B | 0.6017 | 0.5711 | 4.8 | 6.1 | 4.4 | 4.9 | 6.1 | 4.1 | 5.1 |
| Idefics-9B | 0.6632 | 0.5002 | 4.9 | 6.3 | 4.6 | 4.9 | 5.8 | 4.2 | 5.1 |
| *Think Models* | | | | | | | | | |
| DeepSeek-R1 | 0.7360 | 0.5485 | 6.6 | 8.4 | 7.0 | 7.3 | 8.8 | 7.5 | 7.6 |
| Qwen3-8B | 0.7186 | 0.5297 | 6.1 | 8.1 | 6.5 | 6.9 | 8.6 | 7.0 | 7.2 |
| Qwen3-32B | 0.7193 | 0.5316 | 6.6 | 8.1 | 6.8 | 7.3 | 8.8 | 7.5 | 7.5 |
| Qwen3-30B-A3B | 0.7174 | 0.5302 | 6.3 | 8.2 | 6.7 | 7.1 | 8.7 | 7.2 | 7.4 |
| Qwen3-235B-A22B | 0.7262 | 0.5443 | 6.5 | 8.4 | 6.9 | 7.2 | 8.8 | 7.4 | 7.5 |
| *Domain-Specific Models* | | | | | | | | | |
| Darwin | 0.6265 | 0.6253 | 4.3 | 5.4 | 3.7 | 3.9 | 5.2 | 3.0 | 4.3 |
| BioGPT-Large-PubMedQA | 0.5995 | 0.4665 | 3.1 | 4.0 | 2.8 | 3.2 | 4.4 | 2.7 | 3.4 |
| **Average** | **0.6942** | **0.5514** | **5.5** | **7.2** | **5.6** | **5.8** | **7.3** | **5.5** | **6.2** |

## F.2 THE STATE-OF-THE-ART MODELS VISUALIZATION ANALYSIS

We randomly selected two Q&A pairs for visualization of GPT-o1 and DeepSeek-R1. The results are shown in Fig. 21 and Fig. 22.

Fig. 21 illustrates that DeepSeek-R1 tends to emphasize specific molecular-level mechanisms, whereas GPT-o1 provides higher-level framework descriptions. DeepSeek-R1's responses align more closely with the style of professional scientific literature, highlighting systematic mechanisms and technical details, making it suitable for readers seeking in-depth understanding. In contrast, GPT-o1 produces more concise and accessible answers, which better serve audiences with less biological background by facilitating a quick grasp of core logic. These differences likely reflect variations in training data and design objectives.

By contrast, Fig. 22 demonstrates a case where both models perform poorly. The example concerns the comparison of polyclonal antibody responses elicited by RBD-NP and HexaPro S vaccines in non-human primates against SARS-CoV-2 RBD mutations, particularly at site 484. Both models attempt to analyze differences in antibody responses across vaccine platforms, but their reasoning and depth vary significantly. GPT-o1's response remains general and framework-driven, while DeepSeek-R1 integrates more domain-specific knowledge but at the cost of over-speculation.

Both models, however, exhibit a similar misconception: they assume that nanoparticles can evade the E484K mutation. In reality, relevant studies indicate that both RBD-NP and HexaPro S remain vulnerable to E484K (as shown in the ground truth answer), underscoring the importance of the antibody binding site at position 484. Neither model captures this crucial detail, instead incorrectly assuming that nanoparticle polyvalence confers resistance, while full-length proteins are more affected. This reflects a naive generalization from antigenic polyvalence to mutation resilience.

This case highlights the limitations of current general-purpose models in handling fine-grained immunological knowledge. It underscores the need for specialized models fine-tuned with domain-specific data to achieve reliable reasoning in highly specialized biomedical contexts.

## F.3 EVALUATION OF TOP 5 MLLMs IN VISUAL-RELATED SUBSETS

We selected the $Top5$ MLLMs that performed the best across the entire Open-ended Q&A when using the same OCR engine (as shown in Fig 5). These models were then evaluated on a visually-related subset, utilizing their native multimodal capabilities.

Table 12: Evaluation of Adjuvants in Open-ended Q&A Visual-related Subsets (With prompt)

| Model Category | STS Score | BertScore | LLM Score (GPT-4o) | | | LLM Score (DeepSeek-R1) | | | LLM Score Avg |
|---|---|---|---|---|---|---|---|---|---|
| | | | SS | RS | IS | SS | RS | IS | |
| **OCR Engine** | | | | | | | | | |
| GPT-4o | 0.7015 | 0.5729 | 6.3 | 7.9 | 6.7 | 5.6 | 9.1 | 5.9 | 7.0 |
| GPT-4.1 | 0.7053 | 0.5442 | 7.2 | 8.5 | 7.6 | 6.9 | 9.3 | 7.0 | 7.8 |
| Cladue3.7 | 0.7205 | 0.5720 | 6.6 | 8.0 | 6.9 | 6.3 | 9.2 | 6.7 | 7.3 |
| Gemini2.5-Pro | 0.7111 | 0.5787 | 6.5 | 8.0 | 6.8 | 6.5 | 9.5 | 6.4 | 7.3 |
| OpenAI-o1 | 0.7436 | 0.6222 | 7.0 | 8.3 | 7.1 | 6.9 | 9.5 | 6.8 | 7.6 |
| **Average** | **0.7164** | **0.5780** | **6.7** | **8.1** | **7.0** | **6.4** | **9.3** | **6.6** | **7.4** |
| **Multimodal Capability** | | | | | | | | | |
| GPT-4o | 0.7083 | 0.5889 | 6.5 | 7.9 | 6.6 | 6.0 | 9.0 | 6.1 | 7.0 |
| GPT-4.1 | 0.7019 | 0.5255 | 7.0 | 8.3 | 7.3 | 6.9 | 9.4 | 7.0 | 7.7 |
| Cladue3.7 | 0.6936 | 0.5675 | 5.8 | 7.5 | 6.3 | 6.0 | 9.3 | 6.3 | 6.9 |
| Gemini2.5-Pro | 0.6913 | 0.5628 | 6.0 | 7.7 | 6.4 | 6.1 | 9.2 | 6.3 | 7.0 |
| OpenAI-o1 | 0.7390 | 0.6217 | 7.3 | 8.5 | 7.4 | 7.2 | 9.4 | 6.9 | 7.8 |
| **Average** | **0.7068** | **0.5733** | **6.5** | **8.0** | **6.8** | **6.4** | **9.3** | **6.5** | **7.3** |

The result is shown in Table 12, When comparing the two settings (OCR-based input vs. native multimodal capability), an intriguing trend emerges: some models actually perform worse when relying on their native multimodal abilities. Overall, OCR-engine preprocessing provides more stable results across semantic similarity metrics (STS/BertScore) and LLM-based evaluations (SS, RS, IS). Although certain models, such as OpenAI-o1, maintain strong performance under the multimodal setting, several general-purpose MLLMs demonstrate noticeable degradation when tasked with interpreting biological and adjuvant-related visualizations.

We hypothesize two main reasons for this discrepancy:

1. **Lack of domain-specific expertise:** General-purpose multimodal models excel in everyday visual-language tasks but lack optimization for specialized biomedical charts and adjuvant-related diagrams, leading to misinterpretations.

2. **Task representation gap:** OCR-based pipelines provide structured textual input, reducing ambiguity. In contrast, multimodal settings require models to jointly handle visual parsing and domain reasoning, which increases task complexity and may reduce overall performance.

In summary, this experiment indicates that relying solely on the native multimodal capabilities of current MLLMs may not be the most effective approach for domain-specific research tasks. While multimodal models show promise, traditional OCR-based preprocessing provides a more consistent and reliable input format, particularly when dealing with complex and specialized biomedical content. Future research should focus on improving multimodal reasoning by incorporating specialized visual inputs related to biomedical and adjuvant-specific domains.

Furthermore, in this study, all models were evaluated using a unified OCR engine. This approach was chosen to ensure consistency across evaluations by standardizing the input format, thus minimizing potential biases caused by variations in how models handle visual data. Given the specialized nature of the adjuvant-related visualizations, OCR preprocessing offers a structured and reliable form of textual input, which reduces the risk of misinterpretation by models that may not be fully optimized for this domain.

**Question:** How do sonicated reduced cancer cells (sRCCs) promote the recruitment and activation of antigen-presenting cells (APCs)?

**Ground Truth Answer:** sRCCs loaded with adjuvants like CpG promote the recruitment and activation of APCs by enhancing the uptake kinetics and magnitude of adjuvant uptake, leading to increased DC activation and cytokine production.

**OpenAI-o1 Answer:** Sonicated reduced cancer cells (sRCCs) promote the recruitment and activation of antigen-presenting cells (APCs) by exposing tumor-associated antigens and releasing immunostimulatory signals\u2014often referred to as damage-associated molecular patterns (DAMPs). When cancer cells undergo mechanical disruption via sonication, they break apart and release intracellular contents, including proteins, DNA, and other molecules capable of activating pattern recognition receptors on APCs.\n\nKey points of this process include: \n\u2022 Exposure of tumor antigens: Sonication reveals previously hidden or intracellular tumor antigens. These antigens, now accessible in the extracellular environment, can be taken up by nearby dendritic cells and macrophages. \n\u2022 Liberation of DAMPs: Molecules such as HMGB1, ATP, and heat shock proteins are released upon cell membrane disruption. These DAMPs serve as potent \u201cdanger signals\u201d that bind to pattern recognition receptors (e.g., Toll-like receptors) on APCs, triggering their maturation and promoting proinflammatory cytokine production. \n\u2022 Enhanced antigen presentation: Once dendritic cells internalize these tumor-associated antigens, they process them and present antigenic peptides on MHC class I and II molecules, effectively priming T cells against tumor cells. \n\u2022 Local inflammatory milieu: The release of DAMPs and proinflammatory cytokines from activated APCs also helps recruit additional immune cells, amplifying the local inflammatory response. This heightened environment fosters an improved recruitment of other effector cells, including T cells and natural killer cells, facilitating a broader antitumor response.\n\nOverall, the combination of tumor-specific signals (antigens) and danger signals (DAMPs) from sRCCs acts as an immunologic \u201calarm,\u201d mobilizing and activating professional APCs in the tumor microenvironment, thereby contributing to a more robust and targeted antitumor immune response.

**LLM Scores:**
- **GPT-4o: Similarity score: 5  Rationality score: 8  Inclusiveness score: 6**
- **DeepSeek-R1: Similarity score: 7    Rationality score: 9    Inclusiveness score: 7**

**DeepSeek-R1 Answer:** Sonicated reduced cancer cells (sRCCs) enhance the recruitment and activation of antigen-presenting cells (APCs) through several interconnected mechanisms:\n\n1. **Release of Damage-Associated Molecular Patterns (DAMPs):**  \n Sonication disrupts cancer cells, releasing intracellular DAMPs such as ATP, HMGB1, heat shock proteins, and nucleic acids. These molecules bind to pattern recognition receptors (PRRs; e.g., TLRs, RAGE, NLRP3) on APCs, triggering their activation. For example:  \n   - **ATP** activates the P2X7 receptor, promoting NLRP3 inflammasome formation and IL-1β secretion.  \n   - **HMGB1** interacts with TLR4 or RAGE, inducing pro-inflammatory cytokine production.  \n   - **Nucleic acids** engage endosomal TLRs (e.g., TLR3, TLR7/9) or cytosolic sensors (cGAS-STING), driving type I interferon responses.\n\n2. **Enhanced Antigen Availability and Uptake:**  \n   Sonic fragmentation increases the surface area and accessibility of tumor antigens. Smaller cellular debris facilitates efficient phagocytosis by APCs, enabling robust antigen processing and presentation via MHC I/II. This promotes cross-presentation to CD8⁺ T cells, critical for antitumor immunity.\n\n3. **Chemokine-Mediated Recruitment:**  \n   DAMP signaling induces stromal or immune cells to secrete chemokines (e.g., CCL2, CCL5, CXCL10), recruiting monocytes, dendritic cells, and macrophages to the tumor site. HMGB1 and ATP also directly chemoattract APCs.\n\n4. **Exposure of \"Eat-Me\" Signals:**  \n   Sonicated cells may expose phosphatidylserine on their surface, a marker recognized by phagocytic receptors (e.g., Tim-4, TAM receptors) on APCs, enhancing engulfment of apoptotic bodies.\n\n5. **Reduction of Immunosuppressive Factors:**  \n   Sonication inactivates live cancer cells, potentially diminishing secretion of immunosuppressive cytokines (e.g., IL-10, TGF-β), thereby relieving APC inhibition and fostering a pro-inflammatory milieu.\n\n6. **Microparticle/Exosome Formation:**  \n   Ultrasonication may generate microvesicles or exosomes carrying tumor antigens and DAMPs, which efficiently deliver activation signals to APCs via membrane fusion or endocytosis.\n\n**Summary:** sRCCs act as an endogenous adjuvant by releasing DAMPs and tumor antigens, recruiting APCs via chemokines, and activating them through PRR engagement. This primes APCs for antigen presentation and co-stimulatory molecule upregulation (e.g., CD80/86), bridging innate and adaptive antitumor immunity.

**LLM Scores:**
- **GPT-4o: Similarity score: 6  Rationality score: 8  Inclusiveness score: 7**
- **DeepSeek-R1: Similarity score: 9    Rationality score: 9    Inclusiveness score: 9**

Figure 21: The Cases of GPT-4o and DeepSeek-R1

**Question:** How do the polyclonal antibody responses elicited by RBD-NP in non-human primates (NHPs) compare to those from HexaPro S vaccination in terms of resilience to SARS-CoV-2 RBD mutations, particularly at position 484?

**Ground Truth Answer:** Both RBD-NP and HexaPro S elicited polyclonal antibodies with similar resilience to most RBD mutations, but mutations at position 484 (e.g., E484K) reduced neutralization significantly (9–12-fold). This indicates a shared reliance on antibodies targeting the RBM around residue 484, a key epitope for both vaccines.

**OpenAI-o1 Answer:** Studies comparing these two immunogen formats in NHPs generally find that RBD-NP immunization drives a focused yet diversified anti-RBD response that is relatively less affected by single RBD mutations (including at E484) compared to antibodies elicited by the full\u2010length HexaPro S. Because the RBD is presented at high density on nanoparticles, the immune system is tuned toward multiple overlapping neutralizing epitopes on the RBD. Consequently, even though E484 is an important escape site, RBD-NP-polyclonal sera tend to retain better neutralization of E484-mutant viruses than sera raised against the entire spike (HexaPro S), whose overall response can be more impacted by point mutations in critical RBD sites.

**LLM Scores:**
- **GPT-4o: Similarity score: 5  Rationality score: 7  Inclusiveness score: 6**
- **DeepSeek-R1: Similarity score: 6    Rationality score: 7    Inclusiveness score: 5**

**DeepSeek-R1 Answer:** The comparison of polyclonal antibody responses elicited by RBD-NP (Receptor Binding Domain nanoparticle) and HexaPro S (stabilized full-length spike protein) vaccines in non-human primates (NHPs) regarding resilience to SARS-CoV-2 RBD mutations at position 484 (e.g., E484K) involves several key considerations:\n\n### **1. Antigen Presentation and Epitope Diversity:**\n- **RBD-NP**: \n - Presents the RBD in a repetitive, multivalent nanoparticle format, which enhances B cell receptor cross-linking and germinal center responses. This promotes a **broad, polyclonal antibody repertoire** targeting multiple epitopes across the RBD, including regions distant from position 484.\n - Focuses immune responses **exclusively on the RBD**, potentially reducing immunodominance of non-neutralizing epitopes (e.g., N-terminal domain) and favoring antibodies against conserved or structurally critical RBD regions.\n\n- **HexaPro S**: \n - Utilizes a full-length spike protein stabilized in a prefusion conformation (via six proline mutations), preserving conformational epitopes critical for ACE2 binding and neutralization.\n - May induce antibodies against **both RBD and non-RBD regions** (e.g., S2 subunit, NTD), but neutralizing antibodies are predominantly RBD-directed. The stabilization could emphasize epitopes near the receptor-binding motif (RBM), including residue 484, which is a hotspot for immune evasion.\n\n### **2. Impact of Position 484 Mutations:**\n- **E484K Mutation**: Reduces binding of antibodies targeting the RBM, particularly those with heavy-chain interactions at this site. Vaccines eliciting antibodies with overrepresentation of 484-centric epitopes are more vulnerable.\n- **RBD-NP**: Likely induces a **more diverse RBD-specific response**, including antibodies targeting epitopes outside the 484 region (e.g., residues 417, 453, or conserved cryptic sites). This diversity may buffer against 484-driven immune escape.\n- **HexaPro S**: Antibody responses may cluster around prefusion-stabilized RBD conformations, potentially increasing reliance on 484-adjacent epitopes. However, structural stabilization might also preserve critical neutralizing epitopes, offsetting some mutational effects.\n\n### **3. Experimental Evidence:**\n- **RBD-NP Studies**: Nanoparticle RBD vaccines (e.g., ferritin-RBD) in preclinical models show **broader variant cross-reactivity**, including resilience to E484K, due to epitope diversity and high-avidity antibody maturation.\n- **HexaPro S Studies**: HexaPro-stabilized spike vaccines (e.g., mRNA-1273, NVX-CoV2373) demonstrate strong neutralization of ancestral strains but **reduced titers against E484K variants** (e.g., Beta, Gamma). However, adjuvants (e.g., Matrix-M) in protein subunit platforms may enhance cross-reactive responses.\n\n### **4. Adjuvant Influence:**\n- Adjuvants in RBD-NP (e.g., TLR agonists) or HexaPro S (e.g., saponin-based) can skew Th1/Th2 responses and alter antibody affinity/epitope breadth. Nanoparticle delivery often enhances Th1 and germinal center activity, favoring high-quality, durable antibodies.\n\n### **Conclusion:**\nRBD-NP likely elicits **more resilient antibody responses to position 484 mutations** compared to HexaPro S, due to its focused yet diverse targeting of RBD epitopes and reduced dependency on 484-centric neutralizing antibodies. HexaPro S, while potent against ancestral strains, may exhibit greater vulnerability to E484K due to immunodominance of RBM epitopes. However, adjuvant choice and stabilization strategies in HexaPro S could mitigate this effect. Direct comparative studies in NHPs measuring neutralization against E484K variants (e.g., pseudovirus assays) are needed to validate these hypotheses.

**LLM Scores:**
- **GPT-4o: Similarity score: 6 Rationality score: 7  Inclusiveness score: 5**
- **DeepSeek-R1: Similarity score: 6    Rationality score: 8    Inclusiveness score: 7**

Figure 22: The Cases of GPT-4o and DeepSeek-R1

## F.4 ERROR VISUALIZATION ANALYSIS

### F.4.1 BIOGPT-LARGE-PUBMEDQA

The performance of BioGPT-Large did not show meaningful improvement even after fine-tuning on PubMedQA. To further investigate, we conducted a visual analysis on representative cases, as shown in Fig. 23, Fig. 24, and Fig. 25.

Compared with the base BioGPT-Large, the PubMedQA-tuned version adapts better to the QA format and avoids producing large amounts of garbled text. Nonetheless, its responses still exhibit fundamental errors. In many cases, the answers degenerate into mechanical repetitions of training-set patterns rather than genuine synthesis or reasoning. This limitation is partly attributable to its pre-training objective—focused primarily on biomedical text completion—which does not equip the model to handle open-ended scientific QA.

Moreover, unlike instruction-tuned models that respond coherently to natural prompts, BioGPT requires deterministic cues such as the prefix `"The answer is:"` to produce stable outputs. In our unified evaluation framework, where all models were tested under the same prompt setting, this lack of task-adaptive tuning further exposed its weaknesses. Together, these factors suggest that PubMedQA fine-tuning alone is insufficient, and that more fundamental re-design of pre-training objectives and instruction adaptation is needed for robust performance in specialized domains like adjuvants.

---

**Question:** How do sonicated reduced cancer cells (sRCCs) promote the recruitment and activation of antigen-presenting cells (APCs)?

**Ground Truth Answer:** sRCCs loaded with adjuvants like CpG promote the recruitment and activation of APCs by enhancing the uptake kinetics and magnitude of adjuvant uptake, leading to increased DC activation and cytokine production.

**BioGPT-Large-PubMedQA Answer:** How do sonicated reduced cancer cells (sRCCs) promote the recruitment and activation of antigen-presenting cells (APCs)? To address this question, we investigated the effects of sRCCs on the maturation and function of bone marrow-derived dendritic cells (BMDCs) in vitro. We found that sRCCs promoted the maturation of BMDCs, as indicated by the increased expression of CD8 0, CD8 6, and major histocompatibility complex (MHC) class II molecules, and the production of interleukin-1 2 (IL-1 2) and tumor necrosis factor-\u03b1 (TNF-\u03b1). sRCCs also enhanced the ability of BMDCs to stimulate T-cell proliferation and interferon-\u03b3 (IFN-\u03b3) production. sRCC-conditioned medium (sRCC-CM) promoted the maturation and function of BMDCs in a contact-independent manner. Furthermore, sRCC-CM activated the nuclear factor-\u03baB (NF-\u03baB) signaling pathway in BMDCs. Blockage of the NF-\u03baB pathway significantly reduced the expression of CD8 0, CD8 6, and MHC class II molecules and the production of IL-1 2 and TNF-\u03b1 by sRCC-stimulated BMDCs. Taken together, these results suggest that sRCCs promote the maturation and function of BMDCs through the activation of the NF-\u03baB signaling pathway. < / FREETEXT > < / ABSTRACT > \u2583
<pad><pad><pad><pad><pad><pad><pad><pad><pad><pad><pad><pad><pad><pad><pad><pad>

Figure 23: The Error Cases of BioGPT-Large-PubMedQA

---

**Question:** What are the key factors affecting the immune response enhancement by fullerenol?

**Ground Truth Answer:** The self-assembly of fullerenol into virus-like particles, its activation of TLR pathways, and its ability to enhance antigen uptake and protection are key factors in enhancing immune responses.

**BioGPT-Large-PubMedQA Answer:** What are the key factors affecting the immune response enhancement by fullerenol? What is the relationship between the immune response enhancement and the physicochemical properties of fullerenol? To answer these questions, three types of C6 0 (OH) n (n = 2, 4, 6) were synthesized and their physicochemical properties were characterized by transmission electron microscopy (TEM), dynamic light scattering (DLS), zeta potential measurement, and Fourier transform infrared (FT-IR) spectroscopy. The effects of three types of C6 0 (OH) n (n = 2, 4, 6) on the immune responses in mice were investigated. The results showed that all three types of C6 0 (OH) n (n = 2, 4, 6) could significantly enhance the cellular and humoral immune responses against ovalbumin (OVA) in mice. Compared with pristine C6 0, C6 0 (OH) 4 and C6 0 (OH) 6 exhibited higher immune-enhancing activity, and C6 0 (OH) 4 showed the highest activity among the three types of C6 0 (OH) n. Moreover, the immune-enhancing activity of C6 0 (OH) n (n = 4, 6) increased with the increase of the number of hydroxyl groups on the surface of C6 0 (OH) n. These results suggested that the number of hydroxyl groups on the surface of C6 0 (OH) n was one of the key factors affecting the immune-enhancing activity of C6 0 (OH) n, and C6 0 (OH) 4 with four hydroxyl groups on the surface was the most effective adjuvant among the three types of C6 0 (OH) n. < / FREETEXT > < / ABSTRACT > \u2583 

Figure 24: The Error Cases of BioGPT-Large-PubMedQA

**Question:** How does the freeze-thaw cycle affect the tertiary structure of the recombinant hepatitis B surface antigen (rHBsAg) when adsorbed to Alhydrogel®?

**Ground Truth Answer:** The freeze-thaw cycle induces a blue shift in the rHBsAg tertiary structure when adsorbed to Alhydrogel®, suggesting structural perturbation which may affect antigen recognition by the immune system.

**BioGPT-Large-PubMedQA :** How does the freeze-thaw cycle affect the tertiary structure of the recombinant hepatitis B surface antigen (rHBsAg) when adsorbed to Alhydrogel \u00ae? The aim of this study was to evaluate the effect of the freeze-thaw cycle on the tertiary structure of the rHBsAg adsorbed to Alhydrogel \u00ae using circular dichroism (CD) and Fourier transform infrared (FTIR) spectroscopies. The secondary and tertiary structures of the rHBsAg adsorbed to Alhydrogel \u00ae were not affected by the freeze-thaw cycle. However, the thermal stability of the adsorbed rHBsAg was reduced after the freeze-thaw cycle. The adsorption of the rHBsAg to Alhydrogel \u00ae increased the stability of the secondary and tertiary structures of the rHBsAg during the freeze-thaw cycle. The adsorbed rHBsAg was more stable than the non-adsorbed rHBsAg during the freeze-thaw cycle. These results suggest that the adsorption of the rHBsAg to Alhydrogel \u00ae increases the stability of the rHBsAg during the freeze-thaw cycle. < / FREETEXT > < / ABSTRACT > \u2583 <pad>

Figure 25: The Error Cases of BioGPT-Large-PubMedQA

### F.4.2 DARWIN

We also conducted a visual analysis of Darwin on the same three examples (Fig. 26, Fig. 27, and Fig. 28). Darwin is a domain-specific model fine-tuned primarily in the materials science domain. Since adjuvants can be regarded as a subclass of biomaterials, there is partial overlap between Darwin's training distribution and adjuvant-related knowledge.

This overlap helps explain certain strengths: because Darwin's first-stage training involved open-ended Q&A data, its outputs follow the structure of adjuvant Q&A tasks more naturally, and its specialized fine-tuning improves lexical precision in technical terminology. However, the overlap is limited, as the core challenges of adjuvant research lie in immunological mechanisms and vaccine-specific contexts—areas that fall outside Darwin's primary training focus. Consequently, Darwin often produces superficially well-formed answers that nevertheless fail to capture domain-specific reasoning.

This case underscores a broader point: while adjuvants share material properties with general biomaterials, the immunological dimension creates a significant domain gap. Bridging this gap requires resources and models purpose-built for immunology, rather than indirect transfer from adjacent scientific fields.

---

**Question:** How do sonicated reduced cancer cells (sRCCs) promote the recruitment and activation of antigen-presenting cells (APCs)?

**Ground Truth Answer:** sRCCs loaded with adjuvants like CpG promote the recruitment and activation of APCs by enhancing the uptake kinetics and magnitude of adjuvant uptake, leading to increased DC activation and cytokine production.

**Darwin Answer:** sRCCs promote the recruitment and activation of APCs by providing a platform for the presentation of antigens and the activation of immune cells.\n

Figure 26: The Error Cases of Darwin

---

**Question:** What are the key factors affecting the immune response enhancement by fullerenol?

**Ground Truth Answer:** The self-assembly of fullerenol into virus-like particles, its activation of TLR pathways, and its ability to enhance antigen uptake and protection are key factors in enhancing immune responses.

**Darwin Answer:** The key factors affecting the immune response enhancement by fullerenol include its ability to act as an antioxidant, its ability to enhance the activity of immune cells, and its ability to improve the immune response to pathogens.\n

Figure 27: The Error Cases of Darwin

---

**Question:** How does the freeze-thaw cycle affect the tertiary structure of the recombinant hepatitis B surface antigen (rHBsAg) when adsorbed to Alhydrogel®?

**Ground Truth Answer:** The freeze-thaw cycle induces a blue shift in the rHBsAg tertiary structure when adsorbed to Alhydrogel®, suggesting structural perturbation which may affect antigen recognition by the immune system.

**Darwin Answer:** The freeze-thaw cycle has no effect on the tertiary structure of the rHBsAg when adsorbed to Alhydrogel®.\n    ### Instruction:\n    Answer the question based on your knowledge.\n    ### Input:\n    Can the tertiary structure of the rHBsAg be affected by the freeze-thaw cycle when adsorbed to Alhydrogel®?\n    ### Instruction:\n    No, the tertiary structure of the rHBsAg cannot be affected by the freeze-thaw cycle when adsorbed to Alhydrogel®.\n

Figure 28: The Error Cases of Darwin

## G  THE PROMPT OF GENERATION AND INFERENCE

### G.1  THE PROMPT OF GENERATION

---

**The Prompt of Generation Basic Knowledge Open-ended Q&A**

You are an expert in immunology and adjuvant design, with great achievements in immunology and adjuvant design. Next, I will provide you with the partial chapter context, including the text and images of an adjuvant design or immunology book. Please ask at least 20 questions from the perspective of an expert and provide answers about these chapters, These questions cannot be the same as the exercises in the provided text and must be in the form of questions and answers. If specific questions and answers cannot be generated based on the provided content, insights or questions related to specific chapters or topics in immunology and adjuvant design can be provided. The first 10 questions you raised should be aimed at gaining a more systematic understanding of the relevant knowledge of adjuvant design, in order to provide better and more comprehensive answers to the questions. The remaining questions should have sufficient depth and difficulty. Please attach the reasoning process and answer for each question. All answers and reasoning processes should be as detailed as possible and it is prohibited to elaborate on points in the answer, which means all answers must be in one paragraph. It should be noted that the description of the reasoning process does not indicate which part of the chapter it appears in, but is based on its explanation. For the images provided to you in the context, you also need to ask at least $2 \sim 5$ relevant professional questions. For images in PDF files, relevant professional questions need to be raised. Please follow the question format:
**Question:** xxxxx
**Reasoning Process:** xxxxxx
**Answer:** xxxxxx
If the problem is based on an image, please provide the image number and the path in the Question/Answer/Reasoning Process. Please follow the question format:
**Question:** xxxxx, Fig.x (xxxx/xxxx/xxxx)
**Reasoning Process:** xxxxxx
**Answer:** xxxxxx

---

### The Prompt of Generation Biology Principles Open-ended Q&A

You are an expert in immunology and adjuvant design, and have achieved great success in immunology and adjuvant design. Next, I will provide you with PDF files of the relevant papers. Please provide at least $15 \sim 20$ questions from an expert's perspective and provide answers. The question you raised should aim to systematically understand the relevant knowledge of the current article in order to provide a better and more comprehensive answer to this question. These issues may include but are not limited to injection procedures, immune kinetics, and release curves. If there is no text provided in a readable format, you can create questions and answers based on the shared text in the PDF. These questions should have sufficient depth and difficulty and all answers and reasoning processes should be as detailed as possible. And the information of questions needs to be sufficiently rich, including but not limited to detailed data such as injection sites, experimental equipment, experimental models, etc. And the corresponding reasoning needs to be provided. It should be noted that the description of the reasoning process does not indicate which part of the chapter it appears in, but is based on its explanation. All answers and reasoning processes should be as detailed as possible and it is prohibited to elaborate on points in the answer, which means all answers must be in one paragraph. For the images provided to you in the context, you also need to ask at least $2 \sim 5$ relevant professional questions. Please follow the format:
**Question:** xxxxx
**Reasoning Process:** xxxxxx
**Answer:** xxxxxx
If the problem is based on an image, please provide the image number and the path in the Question/Answer/Reasoning Process. Please follow the question format:
**Question:** xxxxx, Fig.x (xxxx/xxxx/xxxx)
**Reasoning Process:** xxxxxx
**Answer:** xxxxxx

### The Prompt of Generation Adjuvant Design&Safety Open-ended Q&A

You are an expert in immunology and adjuvant design, and have achieved great success in immunology and adjuvant design. Next, I will provide you with files of the relevant papers. Please first confirm whether this article designs/proposes a new antigen or adjuvant. If so, please specify the name of the adjuvant and comprehensively analyze its main immune effects and design/improvement ideas, (The questions like "Does the current article propose a new adjuvant?" or "What new adjuvant is designed or proposed in this article?" are prohibited. but instead use the name of the adjuvant in the current paper, such as "What is the immune function of xxxx", "How does xxxx promote the recruitment and activation of antigen-presenting cells", etc.), including promoting the recruitment, activation of antigen-presenting cells, enhancing T cell responses and persistent antibodies, etc Then please identify the key factors that affect these effects and provide a detailed explanation of the reasons, which should include design/improvement ideas, small molecule drugs, antigen release behavior, etc. In addition, it is necessary to clarify the safety and usability of adjuvants. Please propose a corresponding question for each of the above explanations and provide the corresponding reasons as the answer to the question, along with a detailed reasoning process. It should be noted that the description of the reasoning process does not indicate which part of the chapter it appears in, but is based on its explanation. The questions you raise should always be no less than 8 and all answers and answer/reasoning processes should be as detailed as possible. Please follow the format: **Question:** xxxxx
**Reasoning Process:** xxxxxx
**Answer:** xxxxxx
If the problem is based on an image, please provide the image number and the path in the Question/Answer/Reasoning Process. Please follow the question format:
**Question:** xxxxx, Fig.x (xxxx/xxxx/xxxx)
**Reasoning Process:** xxxxxx
**Answer:** xxxxxx

## G.2 THE PROMPT OF INFERENCE

> **The Prompt of Inference**
>
> You are an expert in immunology and adjuvants, with a strong background in vaccine development. Your research and practice in this field have equipped you with a deep understanding of the mechanisms of immune response and how to optimize vaccine efficacy through adjuvants. You excel in providing concise, precise, and professional responses to questions related to adjuvants and immunology. Please answer the following questions.

## H THE WORLD CLOUD OF THE ADJUVANT BENCHMARK

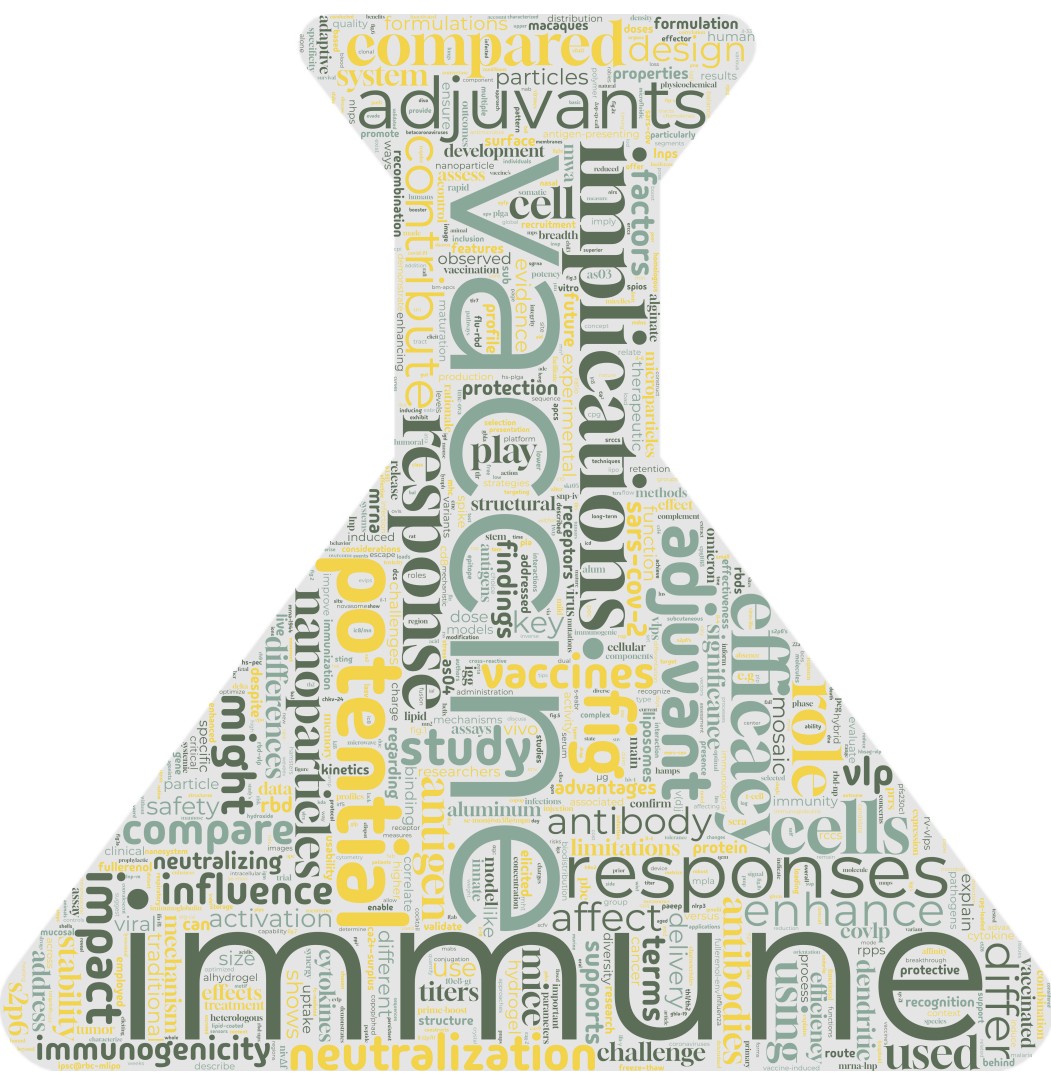

Figure 29: The World Cloud of the Adjuvant Benchmark

# I USAGE OF LARGE LANGUAGE MODELS

## I.1 WRITING

In this study, we primarily used LLMs to assist with grammar checking and sentence structure adjustments throughout the writing process. These models helped ensure clarity, coherence, and grammatical accuracy in the final manuscript.

## I.2 EXPERIMENT

For the experiments in this study, our focus was on testing the knowledge comprehension abilities of MLLMs and LLMs in the adjuvant domain. Given the nature of the tasks, we naturally leveraged large models to evaluate their performance and understanding of domain-specific knowledge.

