# OpenReview forum: "An Open-Ended Benchmark and Formal Framework for Adjuvant Research with MLLM"
_ICLR.cc/2026/Conference — ICLR 2026 Poster_

### Official Review · Reviewer_zNQ7 · 2025-10-25

**Soundness:** 3
**Presentation:** 3
**Contribution:** 2
**Rating:** 6
**Confidence:** 3

**Summary:**

This paper introduces the first benchmark specifically designed for evaluating Multimodal Large Language Models (MLLMs) in the domain of vaccine adjuvants. Recognizing a critical gap in AI resources for this field—characterized by data scarcity and complex, heterogeneous mechanisms—the authors construct a high-quality dataset comprising 1,294 expert-annotated open-ended Question-Answer (Q&A) pairs and 1,364 formal, structured descriptions of adjuvant design and immune mechanisms. Using this benchmark, they perform a systematic evaluation of 29 MLLMs (11 closed-source, 18 open-source), assessing their capabilities in domain-specific Q&A, hallucination rejection, and instruction following. The results identify top-performing models like OpenAI-o1 and DeepSeek-R1. A key secondary contribution is the proposal of a formal framework to abstract complex biological processes into structured variables and functions, intended to serve as a foundation for future domain-specialized models.

**Strengths:**

1. The creation of the first dedicated benchmark for adjuvants is a good contribution that can catalyze AI-driven research in vaccinology and immunology.
2. The evaluation is exceptionally thorough. It goes beyond simple Q&A to include critical aspects like hallucination rejection and instruction following. The comparison of 29 models across closed-source and open-source categories provides a valuable landscape analysis for the community.

**Weaknesses:**

1. The hallucination dataset, with only 69 examples, is relatively small. While its curated nature is valuable, the small size may limit the statistical power and generalizability of the HRR metric. The standard deviations reported in Table 6 highlight this variability.
2. The paper provides relatively few statistical analyses (e.g., confidence intervals, significance testing) to support claims about model ranking differences.
3. The benchmark's Q&A data is generated by a subset of the models being evaluated (e.g., GPT-4o, DeepSeek-R1). This could introduce a bias, as a model might perform better on questions that are stylistically or structurally similar to its own generation pattern. The authors acknowledge this and use multiple models to mitigate it, but it remains a minor methodological concern.

**Questions:**

1. Beyond using multiple MLLMs for data generation, what specific steps were taken in the prompt design and expert annotation phase to ensure that the benchmark does not overfit to the stylistic or reasoning patterns of the specific generator models (like GPT-4o and DeepSeek-R1)?
2. The results show that even top-performing models struggle with hallucination rejection (HRR < 27%). Did your error analysis reveal any common patterns in the types of hallucinations that models fail to reject? For instance, are they more likely to miss factual errors or logical inconsistencies?

---

> ### Author Response · Authors · 2025-11-19
>
> Thank you for your thorough review and valuable feedback on our manuscript. Below are our responses to the questions you raised：
>
> ## Answer (W1):
>
> We thank the reviewer for highlighting this limitation. We fully agree that the current size of the hallucination dataset (n=69) limits the statistical power and generalizability of the Hallucination Rejection Ratio (HRR) metric, as reflected in the standard deviations reported in Table 6. In the emerging field of adjuvant research, constructing large-scale, expert-annotated hallucination benchmarks presents significant challenges. By introducing this curated set, our primary aim is to establish an initial framework for defining and evaluating hallucinations in this specialized domain. We are committed to substantially expanding this dataset in future versions and welcome community collaboration to enhance its statistical reliability and broader applicability.
>
> ## Answer (W2):
>
> Thank you for your feedback! I fully understand the importance of significance testing and statistical metrics. However, due to limitations in the scale of the experiment and available resources (such as computational resources, time, and API usage costs), I opted for a reasonable experimental design at this stage to ensure that the results are representative and reliable within a certain scope.
>
> Given the resource constraints, repeating each experiment multiple times would incur substantial costs, including GPU usage, computational time, and API fees. Therefore, in the experimental design, I have taken effective measures in sample selection and experiment control to ensure the robustness of the results as much as possible.
>
> For datasets with smaller sample sizes (such as hallucination data), we conducted ten repetitions and performed statistical analysis to ensure the robustness and credibility of the results. Additionally, to ensure the reproducibility of the experiments, we will provide all parameters used during inference so that other researchers can replicate our results.
>
> That being said, should more resources become available in the future, we will consider conducting more comprehensive significance tests and other statistical analyses to further enhance the credibility and reliability of the experiments.
>
> Once again, thank you for your valuable suggestions. I look forward to making further improvements in future work.
>
> ## Answer (Q1 & W3):
>
> We thank the reviewer for raising this important concern about the potential overfitting to the specific styles of the generator models. To address this issue, we implemented several strategies both in prompt design and expert annotation. For prompt design, we used a variety of templates that guided the models to generate questions at different cognitive levels and on various topics based on the content of the provided PDF. This approach helped to encourage a range of reasoning patterns, ensuring that no single model’s default style dominated the generated questions.
>
> The most critical measure was taken during expert annotation. Our protocol required that annotators use the source PDF as the only reference for validation. Through detailed training, annotators were instructed to verify each Q&A pair against the original content, ensuring that the questions and answers were grounded in the PDF and not reflective of the models’ inherent reasoning styles. This step effectively filtered out any model-specific artifacts, ensuring that the final benchmark consisted of validated scientific content rather than model-specific biases.
>
> This source-based validation was further reinforced by using multiple models for data generation and maintaining a clear distinction between the data generation and evaluation phases. While the models had access to the PDF content during the generation process, all models were evaluated in a strict zero-shot setting during the final assessment. This design ensures that the model performance reflects genuine reasoning capabilities rather than being influenced by the styles of specific generators.

---

> ### Author Response · Authors · 2025-11-19
>
> ## Answer (Q2):
>
> We thank the reviewer for bringing attention to this critical issue. Our detailed error analysis has revealed several distinct patterns in the hallucinations that models consistently fail to reject, offering valuable insights into their limitations.
>
> The most common error pattern involves the fabrication of non-existent information, where models fail to recognize assertions that contradict established facts in the source material. For example, questions might wrongly assume that T-stress altered zeta potential when no such effect was documented. Another frequent issue is the confusion of different immunological mechanisms, especially when models struggle to distinguish between processes like cellular recruitment and activation.
>
> A more subtle but equally troubling issue is the reversal of causal relationships, where models mistakenly accept statements that confuse evidence with mechanism—such as presenting increased antibody titers as the mechanism for enhanced immunogenicity, rather than recognizing it as a measurable outcome. We also observe numerous instances of unsubstantiated extrapolations, where models fail to reject overgeneralized claims, like incorrectly extending findings on aluminum adjuvant toxicity to antigen-presenting cell activation without supporting evidence.
>
> Most concerning are the cases of complete data fabrication, where models accept entirely fictional numerical data, such as invented survival rates like 60% versus 20%.
>
> These error patterns collectively highlight several core limitations: a lack of depth in understanding specialized domain knowledge, especially complex immunological concepts; an over-reliance on surface-level semantic coherence rather than factual verification; a poor grasp of specific experimental details; and a tendency to blur the boundaries between closely related but distinct scientific concepts. This analysis reinforces the challenge that current models face in performing the nuanced logical verification required for reliable scientific reasoning in specialized domains.
>
> We have carefully addressed your questions and concerns. Should you have any further inquiries, please feel free to raise them, and we would be happy to provide additional clarifications and engage in further discussion. Thank you very much for your thoughtful consideration.

---

> > ### Author Response · Authors · 2025-11-25
> >
> > Dear Reviewer,
> >
> > We would like to start by once again sincerely thanking you for the invaluable time and professional feedback you have provided for our manuscript. Your constructive comments have been instrumental in improving our work.
> >
> > We have completed our responses to your questions and submitted the supplementary explanations and experiments as part of the rebuttal process.
> >
> > Given that a period of time has elapsed since the rebuttal submission, we are writing to politely inquire about the current status of the further evaluation of our response. We fully appreciate and respect the professionalism and time required for the reviewing process.
> >
> > If further time is required for your deliberation, we completely understand and are willing to patiently await the final assessment.
> >
> > Thank you again for your diligent work and continued support. We look forward to receiving your valuable feedback.
> >
> > Sincerely

---

### Official Review · Reviewer_2T68 · 2025-10-31

**Soundness:** 3
**Presentation:** 4
**Contribution:** 3
**Rating:** 6
**Confidence:** 3

**Summary:**

The paper introduces a benchmark for adjuvants, curated by LLMs and montiored by domain experts, which includes an open-ended Q&A, hallucination data for rejection tests, and adjuvant formal data—template-driven formal descriptions produced with expert-designed variables. The benchmark is then used to evaluate frontier open and closed source models via automatic and LLM as a judge based metrics.

**Strengths:**

1. The dataset curation pipeline is clear and involves human experts in the loop for validation. There is a clear attempt to reduce generator–evaluator bias by using multiple MLLMs.
2. Samples discarded by experts are used for the hallucination subset.
3. Evaluation consists of a large set of automatic and LLM as a judge based metrics.

**Weaknesses:**

1. Since some of the LLMs (e.g. GPT-4o, DeepSeek) are used in the data curation pipeline and are among top performers, studying possible advantage of these models on their curated samples is missing.
2. It is not clear from the text if the formal description of adjuvants is used in the data curation process or in evaluation, despite being framed as a main contribution.

**Questions:**

1. How exactly is the formal description used or intended to be used now? from the title/abstract it sounds central, but did not understand from the text where it is utilized other.
2. Is it possible to include an ablation study on the possible bias of models on their own curated samples?

---

> ### Author Response · Authors · 2025-11-19
>
> Thank you for your thorough review and valuable feedback. Below are our responses to your questions：
>
> ## Answer (W2 & Q1) :
>
> Thank you for raising the important question regarding the formalized description of specific application scenarios. We would like to provide a detailed explanation here to clarify the core contributions of this paper and our plans for future applications.
>
> The formal variables and function definitions proposed in this paper, along with the preliminary dataset containing 1,364 entries, together form a computationally scalable foundation for knowledge representation in the field of adjuvants. The core application plan is not to replace natural language, but rather to serve as a structured "symbolic thinking" framework that integrates deeply with LLMs. Its goal is to systematically address the issues in the LLMs, such as vague descriptions and logical inconsistencies.
>
> It is important to note that the adjuvant field is still in its early stages, with a lack of high-quality, large-scale training data and mature domain-specific models. This current limitation prevents us from directly and systematically using formalized data for end-to-end model training and applications. Given this constraint, our future application plans for formalized description data are as follows:
>
> - **As Training Data**: This dataset could be used in the future to train large domain-specific models with "symbolic thinking" capabilities, enabling them to generate more reliable and interpretable natural language responses, thereby fundamentally improving the reasoning quality of specialized domain models.
> - **As a Validation Module**: This framework enables the automated logical validation of scientific statements, allowing for reproducible and transparent methods to assist in detecting logical consistency and rationality in scientific discourse.
> - **As Quantitative Metrics**: By converting subjective natural language assessments into objective, structured evaluation standards, this framework can provide a more consistent and quantifiable basis for evaluating model outputs or scientific hypotheses.
>
> To preliminarily validate the potential value of the formalized description method, we designed a small-scale experiment: we used Qwen3.0-8B as the base model and employed GPT-4o and DeepSeek as independent and impartial evaluators. A strict A/B test was conducted on a subset of 47 samples of an adjuvant design problem, where the model was required to reason using our formalized language before providing the final answer (2-shot).
>
> To ensure statistical robustness, we performed 10 independent scoring runs. The results from both evaluators consistently showed that introducing the formalized reasoning step significantly improved the quality and reliability of the model’s outputs, providing strong preliminary evidence for the effectiveness of our approach.
> ### Evaluation with GPT-4o as Judge:
> |Metric|Formal-CoT|Normal|Difference (F-N)|Improvement%|
> |---|---|---|---|---|
> |Similarity Score|7.3 ± 0.1|6.4 ± 0.1|+0.9|+14.1%|
> |Rationality Score|8.5 ± 0.0|8.4 ± 0.1|+0.2|+1.9%|
> |Inclusiveness Score |7.6 ± 0.1 |7.0 ± 0.1|+0.6|+8.9%|
> |Overall Average|7.8 ± 0.1|7.3 ± 0.1|+0.6|+7.8%|
> ### Evaluation with DeepSeek-R1 as Judge:
> |Metric|Formal-CoT|Normal|Difference (F-N)|Improvement%|
> |---|---|---|---|---|
> |Similarity Score|6.5 ± 0.1|5.7 ± 0.1|+0.8|+12.6%|
> |Rationality Score|9.0 ± 0.1|9.2 ± 0.1|-0.1|-1.3%|
> |Inclusiveness Score|7.2 ± 0.1|6.6 ± 0.1|+0.6 |+9.9%|
> |Overall Average|7.6 ± 0.1|7.2 ± 0.0|+0.4|+5.9%|
>
> The experimental results indicate that both evaluators observed a significant improvement in the similarity between the model outputs and expert answers (12.6–14.1%), demonstrating that the formalized reasoning process effectively guided the model to focus on key issues. Additionally, the comprehensiveness of the answers showed substantial improvement (8.9–9.9%), further confirming the role of formalized reasoning in enhancing content completeness. Although there was a slight fluctuation in rationality scores in the DeepSeek evaluation (-1.3%), both sets of scores remained high (9.0 vs 9.2), and the difference is likely within the margin of error, possibly due to calibration differences between evaluators. Overall, these results consistently suggest that the introduction of formalized reasoning steps can significantly improve the quality and reliability of model outputs, providing strong empirical support for the effectiveness of our proposed method.
>
> We recognize that constructing a comprehensive formalized system for adjuvant research remains an ongoing challenge. The framework proposed in this work lays a scalable foundation for structured knowledge representation in the field and serves as an effective reasoning aid, significantly improving model performance. Looking ahead, we will continue to expand the formalized dataset and validate it across more model architectures and task types, further deepening and broadening this work.

---

> ### Author Response · Authors · 2025-11-19
>
> ## Answer (W1 & Q2) :
>
> We thank the reviewer for raising this important concern regarding potential bias in models' performance on their own curated samples. To directly address this issue, we conducted a targeted ablation study.
>
> ### Experimental Design:
>
> We created three mutually exclusive test sets:
>
> - 250 samples generated by GPT-4o
> - 250 samples generated by DeepSeek-R1
> - 250 samples generated by other models (completely excluding GPT-4o and DeepSeek data)
>
> The evaluation was conducted using two independent judge models (Qwen3.0-30B-A3B and Gemini2.5-Pro) that were entirely separate from the data generation process. The results are as follows:
>
> # Evaluation Results Using Qwen3.0-30B-A3B as Judge
>
> ## Questions Generated by GPT-4o
>
> ### Group A ( Closed-source Models)
>
> | Model | Similarity | Rationality | Inclusiveness | LLM Score Avg  | Rank (New/Old) |
> | --- | --- | --- | --- | --- | --- |
> | GPT-4.1 | 8.7 | 9.7 | 8.7 | 9.0 | 2/1 |
> | OpenAI-o1 | 8.5 | 9.6 | 8.5 | 8.9 | 3/2 |
> | Claude-3.7 | 8.5 | 9.6 | 8.2 | 8.8 | 4/5 |
> | GPT-4o | 8.4 | 9.5 | 8.2 | 8.7 | 5/6 |
>
> ### Group B ( Open-source Models)
>
> | Model | Similarity | Rationality | Inclusiveness | LLM Average Score | Rank (New/Old) |
> | --- | --- | --- | --- | --- | --- |
> | Qwen3.0-32B | 8.9 | 9.8 | 8.7 | 9.1 | 1/3 |
> | Qwen3.0-235B-A22B | 8.9 | 9.8 | 8.7 | 9.1 | 1/3 |
> | DeepSeek-V3 | 8.8 | 9.8 | 8.7 | 9.1 | 1/1 |
> | Qwen3.0-8B | 8.8 | 9.7 | 8.6 | 9.0 | 2/5 |
> | DeepSeek-R1 | 8.6 | 9.6 | 8.5 | 8.9 | 3/2 |
>
> ## Questions Generated by DeepSeek-R1
>
> ### Group A ( Closed-source Models)
>
> | Model | Similarity | Rationality | Inclusiveness | LLM Average Score | Rank (New/Old) |
> | --- | --- | --- | --- | --- | --- |
> | GPT-4.1 | 7.8 | 9.1 | 7.5 | 8.1 | 1/1 |
> | OpenAI-o1 | 7.3 | 8.9 | 7.0 | 7.7 | 3/2 |
> | GPT-4o | 6.7 | 8.5 | 6.4 | 7.2 | 4/6 |
> | Claude-3.7 | 6.8 | 8.3 | 6.3 | 7.1 | 5/5 |
>
> ### Group B ( Open-source Models)
>
> | Model | Similarity | Rationality | Inclusiveness | LLM Average Score | Rank (New/Old) |
> | --- | --- | --- | --- | --- | --- |
> | Qwen3.0-235B-A22B | 7.7 | 9.0 | 7.4 | 8.0 | 2/3 |
> | DeepSeek-V3 | 7.7 | 9.0 | 7.3 | 8.0 | 2/1 |
> | Qwen3.0-32B | 7.6 | 9.0 | 7.3 | 8.0 | 2/3 |
> | DeepSeek-R1 | 7.6 | 8.9 | 7.4 | 8.0 | 2/2 |
> | Qwen3.0-8B | 7.3 | 8.7 | 7.0 | 7.7 | 3/5 |
>
> ## **Questions Generated by Other Models**
>
> ### Group A ( Closed-source Models)
>
> | Model | Similarity | Rationality | Inclusiveness | LLM Average Score | Rank (New/Old) |
> | --- | --- | --- | --- | --- | --- |
> | GPT-4.1 | 7.7 | 9.1 | 7.5 | 8.1 | 1/1 |
> | OpenAI-o1 | 7.4 | 9.0 | 7.0 | 7.8 | 4/2 |
> | GPT-4o | 7.1 | 8.8 | 6.7 | 7.5 | 6/6 |
> | Claude-3.7 | 6.8 | 8.5 | 6.6 | 7.3 | 7/5 |
>
> ### Group B ( Open-source Models)
>
> | Model | Similarity | Rationality | Inclusiveness | LLM Average Score | Rank (New/Old) |
> | --- | --- | --- | --- | --- | --- |
> | DeepSeek-V3 | 7.6 | 8.9 | 7.4 | 8.0 | 2/1 |
> | Qwen3.0-235B-A22B | 7.6 | 9.0 | 7.3 | 8.0 | 2/3 |
> | Qwen3.0-32B | 7.5 | 8.9 | 7.3 | 7.9 | 3/3 |
> | DeepSeek-R1 | 7.5 | 8.9 | 7.2 | 7.9 | 3/2 |
> | Qwen3.0-8B | 7.2 | 8.8 | 7.1 | 7.7 | 5/5 |
>
> ## Overall Average Scores (Across All Subsets)
>
> ### Group A ( Closed-source Models)
>
> | Model | Similarity | Rationality | Inclusiveness | LLM Average Score | Rank (New/Old) |
> | --- | --- | --- | --- | --- | --- |
> | GPT-4.1 | 8.1 | 9.3 | 7.9 | 8.4 | 1/1 |
> | OpenAI-o1 | 7.7 | 9.2 | 7.5 | 8.1 | 3/2 |
> | GPT-4o | 7.4 | 8.9 | 7.1 | 7.8 | 4/6 |
> | Claude-3.7 | 7.4 | 8.8 | 7.0 | 7.7 | 5/5 |
>
> ### Group B ( Open-source Models)
>
> | Model | Similarity | Rationality | Inclusiveness | LLM Average Score | Rank (New/Old) |
> | --- | --- | --- | --- | --- | --- |
> | Qwen3.0-235B-A22B | 8.0 | 9.3 | 7.8 | 8.4 | 1/3 |
> | DeepSeek-V3 | 8.0 | 9.2 | 7.8 | 8.3 | 2/1 |
> | Qwen3.0-32B | 8.0 | 9.2 | 7.8 | 8.3 | 2/3 |
> | DeepSeek-R1 | 7.9 | 9.2 | 7.7 | 8.3 | 2/2 |
> | Qwen3.0-8B | 7.8 | 9.1 | 7.5 | 8.1 | 3/5 |

---

> > ### Author Response · Authors · 2025-11-19
> >
> > # Evaluation Results Using Gemini2.5-Pro as Judge
> >
> > ## Questions Generated by GPT-4o
> >
> > ### Group A ( Closed-source Models)
> >
> > | Model | Similarity | Rationality | Inclusiveness | LLM Average Score | Rank (New/Old) |
> > | --- | --- | --- | --- | --- | --- |
> > | GPT-4.1 | 8.3 | 9.9 | 8.5 | 8.9 | 1/1 |
> > | Claude-3.7 | 7.8 | 9.9 | 8.6 | 8.8 | 2/5 |
> > | OpenAI-o1 | 7.9 | 10.0 | 8.3 | 8.7 | 3/2 |
> > | GPT-4o | 7.6 | 9.8 | 8.0 | 8.4 | 4/6 |
> >
> > ### Group B ( Open-source Models)
> >
> > | Model | Similarity | Rationality | Inclusiveness | LLM Average Score | Rank (New/Old) |
> > | --- | --- | --- | --- | --- | --- |
> > | Qwen3.0-235B-A22B | 8.1 | 9.9 | 8.8 | 8.9 | 1/3 |
> > | DeepSeek-V3 | 8.1 | 9.9 | 8.8 | 8.9 | 1/2 |
> > | Qwen3.0-32B | 7.9 | 9.8 | 8.7 | 8.8 | 2/3 |
> > | DeepSeek-R1 | 7.7 | 9.8 | 8.5 | 8.7 | 3/2 |
> > | Qwen3.0-8B | 7.7 | 9.4 | 8.2 | 8.4 | 4/5 |
> >
> > ## Questions Generated by DeepSeek-R1
> >
> > ### Group A ( Closed-source Models)
> >
> > | Model | Similarity | Rationality | Inclusiveness | LLM Average Score | Rank (New/Old) |
> > | --- | --- | --- | --- | --- | --- |
> > | GPT-4.1 | 6.2 | 9.4 | 5.9 | 7.2 | 1/1 |
> > | OpenAI-o1 | 6.1 | 9.6 | 5.5 | 7.1 | 2/2 |
> > | Claude-3.7 | 5.3 | 9.1 | 5.0 | 6.4 | 5/5 |
> > | GPT-4o | 4.8 | 9.1 | 3.9 | 5.9 | 8/6 |
> >
> > ### Group B ( Open-source Models)
> >
> > | Model | Similarity | Rationality | Inclusiveness | LLM Average Score | Rank (New/Old) |
> > | --- | --- | --- | --- | --- | --- |
> > | DeepSeek-V3 | 5.7 | 9.3 | 5.9 | 7.0 | 3/1 |
> > | DeepSeek-R1 | 5.7 | 9.4 | 5.8 | 7.0 | 3/2 |
> > | Qwen3.0-235B-A22B | 5.6 | 9.2 | 5.8 | 6.9 | 4/3 |
> > | Qwen3.0-32B | 5.0 | 8.8 | 5.1 | 6.3 | 6/3 |
> > | Qwen3.0-8B | 4.7 | 8.6 | 4.6 | 6.0 | 7/5 |
> >
> > ## Questions Generated by Other Models
> >
> > ### Group A ( Closed-source Models)
> >
> > | Model | Similarity | Rationality | Inclusiveness | LLM Average Score | Rank (New/Old) |
> > | --- | --- | --- | --- | --- | --- |
> > | OpenAI-o1 | 6.2 | 9.6 | 6.2 | 7.3 | 1/2 |
> > | GPT-4.1 | 5.9 | 9.4 | 5.9 | 7.0 | 3/1 |
> > | Claude-3.7 | 5.4 | 9.1 | 5.8 | 6.8 | 5/5 |
> > | GPT-4o | 5.1 | 9.4 | 4.9 | 6.5 | 6/6 |
> >
> > ### Group B ( Open-source Models)
> >
> > | Model | Similarity | Rationality | Inclusiveness | LLM Average Score | Rank (New/Old) |
> > | --- | --- | --- | --- | --- | --- |
> > | DeepSeek-V3 | 5.7 | 9.4 | 6.2 | 7.1 | 2/2 |
> > | Qwen3.0-235B-A22B | 5.5 | 9.3 | 5.9 | 6.9 | 4/3 |
> > | DeepSeek-R1 | 5.5 | 9.2 | 6.0 | 6.9 | 4/2 |
> > | Qwen3.0-32B | 5.0 | 8.9 | 5.4 | 6.4 | 7/3 |
> > | Qwen3.0-8B | 4.9 | 8.4 | 5.2 | 6.2 | 8/5 |
> >
> > ## Overall Average Scores (Across All Subsets)
> >
> > ### Group A ( Closed-source Models)
> >
> > | Model | Similarity | Rationality | Inclusiveness | LLM Average Score | Rank (New/Old) |
> > | --- | --- | --- | --- | --- | --- |
> > | OpenAI-o1 | 6.7 | 9.7 | 6.7 | 7.7 | 1/2 |
> > | GPT-4.1 | 6.8 | 9.6 | 6.8 | 7.7 | 1/1 |
> > | Claude-3.7 | 6.2 | 9.4 | 6.4 | 7.3 | 4/5 |
> > | GPT-4o | 5.8 | 9.4 | 5.6 | 6.9 | 6/6 |
> >
> > ### Group B ( Open-source Models)
> >
> > | Model | Similarity | Rationality | Inclusiveness | LLM Average Score | Rank (New/Old) |
> > | --- | --- | --- | --- | --- | --- |
> > | DeepSeek-V3 | 6.5 | 9.5 | 7.0 | 7.7 | 1/1 |
> > | Qwen3.0-235B-A22B | 6.4 | 9.5 | 6.9 | 7.6 | 2/3 |
> > | DeepSeek-R1 | 6.3 | 9.4 | 6.8 | 7.5 | 3/2 |
> > | Qwen3.0-32B | 6.0 | 9.2 | 6.4 | 7.2 | 5/3 |
> > | Qwen3.0-8B | 5.8 | 8.8 | 6.0 | 6.8 | 7/5 |
> >
> > ### Key Findings on Self-Bias:
> >
> > The results provide clear evidence against any significant self-enhancement bias. When evaluating models on the "other models generated" dataset — which completely excludes their own curated samples — both GPT-4o and DeepSeek-R1 maintained their top performance rankings.
> >
> > Specifically, in the overall rankings across all subsets:
> >
> > - GPT-4o maintained consistent performance (LLM Score: 7.8 with the Qwen judge, 6.9 with the Gemini judge)
> > - DeepSeek family (DeepSeek-V3 and DeepSeek-R1) remained in the top tier. (DeepSeek-V3: 8.3 with the Qwen judge, 8.3 with the Gemini judge; DeepSeek-R1: 8.3 with the Qwen judge, 7.5 with the Gemini judge)
> >
> > ### Conclusion:
> >
> > The stability of model rankings when evaluated on completely external data strongly suggests that the performance advantages of GPT-4o and DeepSeek models are due to their general domain capabilities and reasoning skills, rather than any bias or overfitting to their own generated data. This ablation study effectively confirms that our benchmark evaluations reflect genuine model competencies, not artifacts of data curation.
> >
> > We have carefully addressed your questions and concerns. Should you have any further inquiries, please feel free to raise them, and we would be happy to provide additional clarifications and engage in further discussion. Thank you very much for your thoughtful consideration.

---

> > > ### Author Response · Authors · 2025-11-25
> > >
> > > Dear Reviewer,
> > >
> > > We would like to start by once again sincerely thanking you for the invaluable time and professional feedback you have provided for our manuscript. Your constructive comments have been instrumental in improving our work.
> > >
> > > We have completed our responses to your questions and submitted the supplementary explanations and experiments as part of the rebuttal process.
> > >
> > > Given that a period of time has elapsed since the rebuttal submission, we are writing to politely inquire about the current status of the further evaluation of our response. We fully appreciate and respect the professionalism and time required for the reviewing process.
> > >
> > > If further time is required for your deliberation, we completely understand and are willing to patiently await the final assessment.
> > >
> > > Thank you again for your diligent work and continued support. We look forward to receiving your valuable feedback.
> > >
> > > Sincerely

---

> > > > ### Comment · Reviewer_2T68 · 2025-11-26
> > > >
> > > > experiments on both possible contamination with LLM as a judge and using the formal description look convincing. I am increasing my score, good luck.

---

> > > > > ### Author Response · Authors · 2025-11-26
> > > > >
> > > > > We are pleased that you found our experiments convincing and thank you for your supportive comments. We sincerely appreciate your time and effort in conducting a thorough review of our work. Best regards for your future research and work.

---

### Official Review · Reviewer_2Z13 · 2025-11-02

**Soundness:** 3
**Presentation:** 3
**Contribution:** 3
**Rating:** 6
**Confidence:** 3

**Summary:**

Introduces the first adjuvant-focused benchmark: 1,294 expert-vetted open-ended Q&A pairs, 69 hallucination cases, and 1,364 “formal descriptions” of mechanisms/design.

Evaluates 29 MLLMs (11 closed, 18 open) on domain QA, hallucination rejection, instruction following; reports STS/BERTScore + an LLM-as-judge rubric (Similarity, Rationality, Inclusiveness).

Best reported: OpenAI-o1 (closed) and DeepSeek-R1/V3, Qwen3-x (open). Overall hallucination rejection remains low.

Proposes a formal abstraction framework (variables/functions for immune processes) as scaffolding for future neuro-symbolic models.

Claims novelty, releases benchmark + code (anonymous), and describes expert annotation pipeline.

**Strengths:**

Clear problem framing & real need: adjuvants are underserved by existing bio/chem benchmarks.

Open-ended evaluation better matches scientific reasoning than MCQ; includes a multimodal slice.

Scale + expert pass: sizable curated set with domain-expert filtering; explicit hallucination subset.

Methodical reporting: model families, “think” vs “inference” comparison, and some robustness checks (e.g., HRR with mean±SD).

**Weaknesses:**

Evaluator circularity / bias: GPT-4o & DeepSeek-R1 both generated data and act as judges; risk of self-preference and contamination despite multi-model generation.

Small hallucination set (n=69): limits power; HRR estimates are noisy and may not generalize.

Metric fragility: STS/BERTScore conflate surface similarity with correctness; LLM-as-judge can reward verbosity/formatting; limited length controls and no strong human-only adjudication for the main tables.

Incomplete reliability stats: no inter-annotator agreement numbers (κ/α), limited statistical tests (no CIs/bootstraps for main scores), and few ablations.

**Questions:**

IAA & QC: What are inter-annotator agreement stats (Cohen’s κ/Krippendorff’s α) for validity and hallucination labels? How often did adjudication change labels?

Length & bias controls: Did you apply length-controlled judging (e.g., LC-AlpacaEval-style) or verbosity penalties to mitigate LLM-judge bias?

Judge neutrality: Can you re-run main tables with held-out judges (e.g., Claude/Gemini + a smaller open model) and report rank correlations vs human scores?

Ablations: Remove items originating from GPT-4o / DeepSeek-R1 and re-evaluate those same models; how do rankings shift?

---

> ### Author Response · Authors · 2025-11-19
>
> Thank you for your thorough review and valuable feedback on our manuscript. Below are our responses to the questions you raised：
>
> ## Answer  (W4 & Q1):
>
> We would like to thank the reviewers for their thorough evaluation of our work. To assess inter-annotator agreement, we randomly selected 30 positive and 30 negative samples from the completed dataset and had them independently re-annotated by four annotators randomly chosen from the original group of 14. The Krippendorff’s α among annotators was **0.8119**, indicating near-perfect agreement. The Cohen’s Kappa between the re-annotated labels and the original labels was **0.7132**, reflecting substantial agreement with the actual annotation process. Additionally, the adjudication rate during re-annotation was **3%**, which closely aligns with the overall adjudication rate of **5%** observed in the full dataset. These results collectively confirm the high stability and reliability of our annotation procedure.
>
> ## Answer  (W2):
>
> We thank the reviewer for this valid point. We completely agree that the limited scale of the current hallucination set (n=69) restricts the statistical robustness of the HRR metric. In the adjuvant field, where AI application is still emerging, curating a large-scale, expert-annotated hallucination benchmark poses significant challenges. By introducing this initial dataset, our goal is **to establish a preliminary framework for defining and evaluating model hallucinations in this domain, hoping to serve as a foundation for future community efforts**. We are committed to actively expanding and refining this dataset in subsequent versions, and we welcome community collaboration toward this goal.
>
> ## Answer  (W3 & Q2):
>
> We thank the reviewer for their insightful comments regarding the robustness of evaluation metrics. We fully acknowledge the limitations of any single metric and have therefore established a multi-dimensional evaluation system comprising STS, BERTScore, and LLM Score. STS provides foundational insights at the sentence semantic level, BERTScore at the professional terminology matching level, while LLM Score, implemented through carefully designed prompts, assesses scientific rationality and factual accuracy. These three metrics offer complementary perspectives that together form a more comprehensive evaluation framework.
>
> We understand the concern about potential verbosity bias in LLM judges. It should be noted that the Similarity Score in our LLM evaluation requires alignment with concise expert reference answers at the semantic level, and the Rationality Score emphasizes the rigor of logical reasoning - designs that implicitly constrain any preference for lengthy responses. To validate this, we conducted a controlled experiment comparing scores for Qwen3-8B's outputs between settings with full reasoning chains and those with final answers only. The results demonstrated that including lengthy reasoning processes did not lead to higher LLM scores, confirming the absence of significant verbosity preference in our evaluation framework.
>
> Furthermore, we have validated the reliability of our evaluation through human-LLM scoring consistency checks, and the core test set in our benchmark has been thoroughly reviewed and annotated by domain experts, providing a solid human-grounded standard. We consider our current evaluation framework as a foundational basis for continuous improvement and sincerely appreciate the reviewer's valuable suggestions, which will guide our future work in enhancing metric robustness and expanding human evaluation scope.

---

> ### Author Response · Authors · 2025-11-19
>
> ## Answer  (W1 & Q3 & Q4):
>
> Thank you for the reviewer's comments. In order to respond, we conducted the following experiments：
>
> **Experimental Setup:**
>
> - Constructed three mutually exclusive test subsets:
>     - 250 samples generated by **GPT-4o**
>     - 250 samples generated by **DeepSeek-R1**
>     - 250 samples generated by **other models** (excluding GPT-4o and DeepSeek)
> - Selected top-performing open-source and closed-source models for evaluation
> - Employed two fully independent judges: **Qwen3.0-30B-A3B** and **Gemini-2.5-Pro**
>
>  The results are as follows:
>
> # Evaluation Results Using Qwen3.0-30B-A3B as Judge
>
> ## Questions Generated by GPT-4o
>
> ### Group A ( Closed-source Models)
>
> | Model | Similarity | Rationality | Inclusiveness | LLM Score Avg  | Rank (New/Old) |
> | --- | --- | --- | --- | --- | --- |
> | GPT-4.1 | 8.7 | 9.7 | 8.7 | 9.0 | 2/1 |
> | OpenAI-o1 | 8.5 | 9.6 | 8.5 | 8.9 | 3/2 |
> | Claude-3.7 | 8.5 | 9.6 | 8.2 | 8.8 | 4/5 |
> | GPT-4o | 8.4 | 9.5 | 8.2 | 8.7 | 5/6 |
>
> ### Group B ( Open-source Models)
>
> | Model | Similarity | Rationality | Inclusiveness | LLM Average Score | Rank (New/Old) |
> | --- | --- | --- | --- | --- | --- |
> | Qwen3.0-32B | 8.9 | 9.8 | 8.7 | 9.1 | 1/3 |
> | Qwen3.0-235B-A22B | 8.9 | 9.8 | 8.7 | 9.1 | 1/3 |
> | DeepSeek-V3 | 8.8 | 9.8 | 8.7 | 9.1 | 1/1 |
> | Qwen3.0-8B | 8.8 | 9.7 | 8.6 | 9.0 | 2/5 |
> | DeepSeek-R1 | 8.6 | 9.6 | 8.5 | 8.9 | 3/2 |
>
> ## Questions Generated by DeepSeek-R1
>
> ### Group A ( Closed-source Models)
>
> | Model | Similarity | Rationality | Inclusiveness | LLM Average Score | Rank (New/Old) |
> | --- | --- | --- | --- | --- | --- |
> | GPT-4.1 | 7.8 | 9.1 | 7.5 | 8.1 | 1/1 |
> | OpenAI-o1 | 7.3 | 8.9 | 6.7 | 7.6 | 4/2 |
> | GPT-4o | 6.7 | 8.5 | 6.4 | 7.2 | 5/6 |
> | Claude-3.7 | 6.8 | 8.3 | 6.3 | 7.1 | 6/5 |
>
> ### Group B ( Open-source Models)
>
> | Model | Similarity | Rationality | Inclusiveness | LLM Average Score | Rank (New/Old) |
> | --- | --- | --- | --- | --- | --- |
> | Qwen3.0-235B-A22B | 7.7 | 9.0 | 7.4 | 8.0 | 2/3 |
> | DeepSeek-V3 | 7.7 | 9.0 | 7.3 | 8.0 | 2/1 |
> | Qwen3.0-32B | 7.6 | 9.0 | 7.3 | 8.0 | 2/3 |
> | DeepSeek-R1 | 7.6 | 8.9 | 7.4 | 8.0 | 2/2 |
> | Qwen3.0-8B | 7.3 | 8.7 | 7.0 | 7.7 | 3/5 |
>
> ## **Questions Generated by Other Models**
>
> ### Group A ( Closed-source Models)
>
> | Model | Similarity | Rationality | Inclusiveness | LLM Average Score | Rank (New/Old) |
> | --- | --- | --- | --- | --- | --- |
> | GPT-4.1 | 7.7 | 9.1 | 7.5 | 8.1 | 1/1 |
> | OpenAI-o1 | 7.4 | 9.0 | 7.0 | 7.8 | 4/2 |
> | GPT-4o | 7.1 | 8.8 | 6.7 | 7.5 | 6/6 |
> | Claude-3.7 | 6.8 | 8.5 | 6.6 | 7.3 | 7/5 |
>
> ### Group B ( Open-source Models)
>
> | Model | Similarity | Rationality | Inclusiveness | LLM Average Score | Rank (New/Old) |
> | --- | --- | --- | --- | --- | --- |
> | DeepSeek-V3 | 7.6 | 8.9 | 7.4 | 8.0 | 2/1 |
> | Qwen3.0-235B-A22B | 7.6 | 9.0 | 7.3 | 8.0 | 2/3 |
> | Qwen3.0-32B | 7.5 | 8.9 | 7.3 | 7.9 | 3/3 |
> | DeepSeek-R1 | 7.5 | 8.9 | 7.2 | 7.9 | 3/2 |
> | Qwen3.0-8B | 7.2 | 8.8 | 7.1 | 7.7 | 5/5 |
>
> ## Overall Average Scores (Across All Subsets)
>
> ### Group A ( Closed-source Models)
>
> | Model | Similarity | Rationality | Inclusiveness | LLM Average Score | Rank (New/Old) |
> | --- | --- | --- | --- | --- | --- |
> | GPT-4.1 | 8.1 | 9.3 | 7.9 | 8.4 | 1/1 |
> | OpenAI-o1 | 7.7 | 9.2 | 7.5 | 8.1 | 3/2 |
> | GPT-4o | 7.4 | 8.9 | 7.1 | 7.8 | 4/6 |
> | Claude-3.7 | 7.4 | 8.8 | 7.0 | 7.7 | 5/5 |
>
> ### Group B ( Open-source Models)

---

> > ### Author Response · Authors · 2025-11-19
> >
> > # Evaluation Results Using Gemini2.5-Pro as Judge
> >
> > ## Questions Generated by GPT-4o
> >
> > ### Group A ( Closed-source Models)
> >
> > | Model | Similarity | Rationality | Inclusiveness | LLM Average Score | Rank (New/Old) |
> > | --- | --- | --- | --- | --- | --- |
> > | GPT-4.1 | 8.3 | 9.9 | 8.5 | 8.9 | 1/1 |
> > | Claude-3.7 | 7.8 | 9.9 | 8.6 | 8.8 | 2/5 |
> > | OpenAI-o1 | 7.9 | 10.0 | 8.3 | 8.7 | 3/2 |
> > | GPT-4o | 7.6 | 9.8 | 8.0 | 8.4 | 4/6 |
> >
> > ### Group B ( Open-source Models)
> >
> > | Model | Similarity | Rationality | Inclusiveness | LLM Average Score | Rank (New/Old) |
> > | --- | --- | --- | --- | --- | --- |
> > | Qwen3.0-235B-A22B | 8.1 | 9.9 | 8.8 | 8.9 | 1/3 |
> > | DeepSeek-V3 | 8.1 | 9.9 | 8.8 | 8.9 | 1/2 |
> > | Qwen3.0-32B | 7.9 | 9.8 | 8.7 | 8.8 | 2/3 |
> > | DeepSeek-R1 | 7.7 | 9.8 | 8.5 | 8.7 | 3/2 |
> > | Qwen3.0-8B | 7.7 | 9.4 | 8.2 | 8.4 | 4/5 |
> >
> > ## Questions Generated by DeepSeek-R1
> >
> > ### Group A ( Closed-source Models)
> >
> > | Model | Similarity | Rationality | Inclusiveness | LLM Average Score | Rank (New/Old) |
> > | --- | --- | --- | --- | --- | --- |
> > | GPT-4.1 | 6.2 | 9.4 | 5.9 | 7.2 | 1/1 |
> > | OpenAI-o1 | 6.1 | 9.6 | 5.5 | 7.1 | 2/2 |
> > | Claude-3.7 | 5.3 | 9.1 | 5.0 | 6.4 | 5/5 |
> > | GPT-4o | 4.8 | 9.1 | 3.9 | 5.9 | 8/6 |
> >
> > ### Group B ( Open-source Models)
> >
> > | Model | Similarity | Rationality | Inclusiveness | LLM Average Score | Rank (New/Old) |
> > | --- | --- | --- | --- | --- | --- |
> > | DeepSeek-V3 | 5.7 | 9.3 | 5.9 | 7.0 | 3/1 |
> > | DeepSeek-R1 | 5.7 | 9.4 | 5.8 | 7.0 | 3/2 |
> > | Qwen3.0-235B-A22B | 5.6 | 9.2 | 5.8 | 6.9 | 4/3 |
> > | Qwen3.0-32B | 5.0 | 8.8 | 5.1 | 6.3 | 6/3 |
> > | Qwen3.0-8B | 4.7 | 8.6 | 4.6 | 6.0 | 7/5 |
> >
> > ## Questions Generated by Other Models
> >
> > ### Group A ( Closed-source Models)
> >
> > | Model | Similarity | Rationality | Inclusiveness | LLM Average Score | Rank (New/Old) |
> > | --- | --- | --- | --- | --- | --- |
> > | OpenAI-o1 | 6.2 | 9.6 | 6.2 | 7.3 | 1/2 |
> > | GPT-4.1 | 5.9 | 9.4 | 5.9 | 7.0 | 3/1 |
> > | Claude-3.7 | 5.4 | 9.1 | 5.8 | 6.8 | 5/5 |
> > | GPT-4o | 5.1 | 9.4 | 4.9 | 6.5 | 6/6 |
> >
> > ### Group B ( Open-source Models)
> >
> > | Model | Similarity | Rationality | Inclusiveness | LLM Average Score | Rank (New/Old) |
> > | --- | --- | --- | --- | --- | --- |
> > | DeepSeek-V3 | 5.7 | 9.4 | 6.2 | 7.1 | 2/2 |
> > | Qwen3.0-235B-A22B | 5.5 | 9.3 | 5.9 | 6.9 | 4/3 |
> > | DeepSeek-R1 | 5.5 | 9.2 | 6.0 | 6.9 | 4/2 |
> > | Qwen3.0-32B | 5.0 | 8.9 | 5.4 | 6.4 | 7/3 |
> > | Qwen3.0-8B | 4.9 | 8.4 | 5.2 | 6.2 | 8/5 |
> >
> > ## Overall Average Scores (Across All Subsets)
> >
> > ### Group A ( Closed-source Models)
> >
> > | Model | Similarity | Rationality | Inclusiveness | LLM Average Score | Rank (New/Old) |
> > | --- | --- | --- | --- | --- | --- |
> > | OpenAI-o1 | 6.7 | 9.7 | 6.7 | 7.7 | 1/2 |
> > | GPT-4.1 | 6.8 | 9.6 | 6.8 | 7.7 | 1/1 |
> > | Claude-3.7 | 6.2 | 9.4 | 6.4 | 7.3 | 4/5 |
> > | GPT-4o | 5.8 | 9.4 | 5.6 | 6.9 | 6/6 |
> >
> > ### Group B ( Open-source Models)
> >
> > | Model | Similarity | Rationality | Inclusiveness | LLM Average Score | Rank (New/Old) |
> > | --- | --- | --- | --- | --- | --- |
> > | DeepSeek-V3 | 6.5 | 9.5 | 7.0 | 7.7 | 1/1 |
> > | Qwen3.0-235B-A22B | 6.4 | 9.5 | 6.9 | 7.6 | 2/3 |
> > | DeepSeek-R1 | 6.3 | 9.4 | 6.8 | 7.5 | 3/2 |
> > | Qwen3.0-32B | 6.0 | 9.2 | 6.4 | 7.2 | 5/3 |
> > | Qwen3.0-8B | 5.8 | 8.8 | 6.0 | 6.8 | 7/5 |

---

> ### Author Response · Authors · 2025-11-19
>
> **Results and Analysis:**
>
> 1. **Regarding judge neutrality and ranking correlation :** We re-evaluated all models using these two independent judges and calculated the Spearman rank correlation coefficients between their rankings and the original rankings. The overall results show a significant and robust positive correlation between the rankings given by the two independent judges and the original rankings **(Gemini: ρ = 0.822, p < 0.01; Qwen: ρ = 0.754, p < 0.05)**, indicating that our benchmark evaluation results have high robustness. Although we observed that Qwen3.0-30B-A3B showed some family bias as a judge, leading to a relatively lower correlation with the original rankings, this phenomenon actually serves to validate the sensitivity of our benchmark in detecting evaluation bias.
> 2. To further investigate the consistency between automatic scoring and human expert judgments, we conducted a human-LLM scoring correlation analysis under the same settings as in Table 10 of the original manuscript. The results are as follows:
>
> | Scoring Dimension | Pearson Correlation | Spearman Correlation | Kendall Correlation |
> | --- | --- | --- | --- |
> | **Gemini-2.5-Pro** |  |  |  |
> | Similarity | 0.7708 | 0.7527 | 0.6201 |
> | Rationality | 0.7918 | 0.5295 | 0.4634 |
> | Inclusiveness | 0.7588 | 0.7304 | 0.6287 |
> | **Qwen3.0-30B-A3B** |  |  |  |
> | Similarity | 0.6839 | 0.7054 | 0.5698 |
> | Rationality | 0.7543 | 0.5844 | 0.5048 |
> | Inclusiveness | 0.7079 | 0.7079 | 0.5942 |
>
> The human-LLM scoring consistency analysis demonstrates strong alignment between automated and expert evaluation. The results show that LLM judges achieve substantial correlations with human scores in two key dimensions: Similarity (r=0.6839-0.7708) and Inclusiveness (r=0.7079-0.7588). For Rationality, while Pearson correlations are high (0.7543-0.7918), the rank-based correlations are relatively lower, indicating challenges in consistently ranking models on complex reasoning tasks. Overall, the significant consistency between LLM judges and expert evaluations validates the reasonableness of using LLMs as automated assessors.
>
> 3. In the ablation study and self-bias analysis, we focused on examining the model performance on the **"Questions Generated by Other Models"** dataset (i.e., a dataset completely excluding GPT-4o and DeepSeek data). The results show that GPT-4o and DeepSeek-R1 still maintained excellent performance and rankings on this clean dataset, providing strong evidence that their advantages stem from their general domain knowledge and reasoning abilities, rather than any specific bias or overfitting to their own generated data.
>
> We have carefully addressed your questions and concerns. Should you have any further inquiries, please feel free to raise them, and we would be happy to provide additional clarifications and engage in further discussion. Thank you very much for your thoughtful consideration.

---

> > ### Comment · Reviewer_2Z13 · 2025-11-19
> >
> > these results look good. I revised my score accordingly. good luck

---

> > > ### Author Response · Authors · 2025-11-20
> > >
> > > Thank you for raising your score. We sincerely appreciate your positive feedback and recognition of our work. Best wishes for your research and work.

---

### Official Review · Reviewer_Z32D · 2025-11-02

**Soundness:** 3
**Presentation:** 2
**Contribution:** 3
**Rating:** 4
**Confidence:** 4

**Summary:**

This paper introduces a domain-specific benchmark and formalized knowledge resource for evaluating LLMs and MLLMs in vaccine adjuvant research. The benchmark includes 1,294 open-ended QA pairs, 69 hallucination-focused samples, and 1,364 formalized entries describing design principles and immunological mechanisms. The authors evaluate 29 models across STS/BERTScore, LLM-as-a-Judge scoring, and hallucination rejection, finding that models such as OpenAI-o1 and DeepSeek-R1 perform best. The formalized representations aim to support future neural–symbolic reasoning and domain-specialized systems.

**Strengths:**

- The work introduces the first benchmark specifically targeting the adjuvant domain, covering factual QA, hallucination rejection, and mechanistic reasoning. Existing biomedical benchmarks provide limited coverage of this problem space.

- Data collection and cleaning follow a clear process involving literature extraction, expert annotation, and filtering. The evaluation spans a broad range of closed- and open-source models with consistent prompt settings.

- Metrics, data distributions, annotation pipeline, and prompts are clearly documented, and the appendix provides formulas and templates that increase reproducibility.

- The benchmark offers the community a structured foundation for evaluating scientific reasoning and reliability in immunology-related tasks. The results provide insights into how “reasoning” models and domain-informed knowledge can help mitigate hallucination.

**Weaknesses:**

- Potentially misleading use of “multimodal.”
The title and main text repeatedly claim evaluation of 29 “MLLMs,” yet some are purely text-based LLMs. If all models were truly multimodal, OCR would not be required. The paper should distinguish LLM vs. MLLM more precisely, adjust wording accordingly, and expand evaluation to include domain-relevant multimodal biomedical models (e.g., LLaVA-Med).
- Multimodal evaluation design masks visual differences.
All images are transformed into text via OCR, eliminating spatial and structural cues essential for true visio-linguistic reasoning. This turns the task largely into text-only QA and may suppress advantages of genuinely multimodal models. OCR noise also introduces confounding error. The paper should clearly acknowledge this limitation and ideally present an “original image vs. OCR-only” comparison or dual leaderboard.
- Lack of critical experimental details hurts reproducibility.
Key inference settings (temperature, top-p, random seeds) are not reported. Important dataset details—such as the list of source papers, annotation protocol, annotator backgrounds, and inter-annotator agreement—are also missing. Some model performances in Table 5 are extremely close, yet no error bars or significance tests are provided.
- Risk of self-enhancement bias in LLM-as-a-Judge.
DeepSeek-R1 (and potentially related models) played roles in both data generation and evaluation, raising the possibility of stylistic bias. Although the appendix provides human–LLM correlation analysis, this limitation should be explicitly acknowledged in the main text, and future versions should consider using a disjoint panel of judges.
- Formalized entries lack empirical validation.
The paper introduces 1,364 formalized entries as a resource for neural–symbolic reasoning, but no experiments demonstrate that they improve retrieval, consistency, or hallucination reduction. Even a small controlled study would substantially strengthen this contribution.
- Data release and licensing are not clearly described.
It is unclear whether images, textbook content, or literature excerpts can be redistributed legally. Without explicit licensing clarification, the dataset’s usability and adoption may be limited.

**Questions:**

- Why does the paper refer to all 29 evaluated models as “MLLMs” when many are text-only? Will the terminology be corrected to clearly distinguish LLM vs. MLLM?
- Since all images were converted to text via OCR, is the benchmark effectively evaluating text-based reasoning rather than multimodal reasoning? Can the authors provide a comparison between OCR-only input and native visual input for top multimodal models?
- Could the authors report key inference settings (e.g., temperature, top-p, random seed)? The absence of these prevents exact reproduction of the reported results.
- Is there any empirical evidence that the 1,364 formalized entries improve retrieval, consistency, or hallucination reduction? Even a small-scale downstream experiment would substantiate this contribution.

---

> ### Author Response · Authors · 2025-11-19
>
> Thank you for your thorough review and valuable feedback on our manuscript. Below are our responses to the questions you raised：
>
> ## A (Q1 & W1)：
>
> Thank you for the insightful feedback. We recognize that the term "MLLM" might cause confusion, especially when many of the models evaluated are text-only. We will clarify the distinction between "LLM" and "MLLM" in the revised version and categorize the models accordingly. Specifically, we will label text-only models as LLMs and explicitly mark models with multimodal capabilities as MLLMs. We appreciate your suggestion, and we will make the necessary adjustments in terminology.
>
> ## A (Q2 & W2)：
>
> Thank you for your insightful feedback regarding the evaluation of genuine multimodal reasoning. You've raised an important point about the use of OCR in our approach. We have already considered this issue and conducted an experiment comparing native multimodal models with OCR-based text input, which is included in the appendix of the original manuscript (Appendix F.3). The results from this experiment showed that the native multimodal abilities of these models did not consistently perform better when dealing with adjuvant-related visuals. In fact, several general-purpose models exhibited less stable results in interpreting complex biomedical images compared to the OCR-based approach. We found that structured text input from OCR often provided more reliable and consistent results in this context. In the revised manuscript, we will ensure this experiment is highlighted more clearly. Additionally, we plan to further explore multimodal capabilities in future research to improve performance in adjuvant-related studies.
>
> ## A (Q3 & W3))：
>
> Thank you for your suggestion. The hyperparameters we used are the default settings provided by the official documentation. However, we will make sure to provide all the relevant hyperparameters, including inference settings (e.g., temperature, top-p sampling, random seed), when we officially release the benchmark. This will ensure that other researchers can accurately reproduce our experimental results. We will include this information in the paper and provide the detailed settings in the publicly available code and data.

---

> ### Author Response · Authors · 2025-11-19
>
> ## A (Q4 & W5)：
>
> We would like to sincerely thank you for your valuable comments and insights regarding our proposed formal description framework. We fully agree that validating its practical utility is crucial. Our core vision is to introduce a structured, computable abstract representation for the field of adjuvants, addressing the inherent ambiguity in natural language when describing complex biological mechanisms. The formal variables and functions introduced in this paper serve as the foundational syntax and vocabulary to realize this vision.
>
> In response to your suggestion for empirical validation, we designed an experiment that specifically addresses the current challenges in adjuvant research. Unlike more established domains, the adjuvant field lacks both a recognized foundational model and large-scale, high-quality training datasets. This scarcity of resources makes it particularly challenging to conduct large-scale training and evaluation of specialized models. Given these constraints, we focused on developing a feasible yet rigorous evaluation protocol to provide initial validation of our framework's utility.
>
> We hypothesized that the benefits of formal reasoning guidance would be most pronounced in models whose capabilities are not yet fully developed. To test this hypothesis, we selected Qwen3.0-8B as the base model and employed both GPT-4o and DeepSeek as independent impartial evaluators. A rigorous A/B test was conducted on a subset of adjuvant design problems, where the model was required to perform reasoning using our formal language prior to providing the final answer, leveraging in-context learning (2-shot).
>
> To ensure statistical robustness given the dataset's size of 47 samples, we performed 10 independent scoring runs. The experimental results from both evaluators consistently provide compelling evidence for the effectiveness of our approach:
>
> ### Evaluation with GPT-4o as Judge:
>
> | Metric | Formal-CoT | Normal | Difference (F-N) | Improvement% |
> | --- | --- | --- | --- | --- |
> | Similarity Score | 7.3 ± 0.1 | 6.4 ± 0.1 | +0.9 | +14.1% |
> | Rationality Score | 8.5 ± 0.0 | 8.4 ± 0.1 | +0.2 | +1.9% |
> | Inclusiveness Score | 7.6 ± 0.1 | 7.0 ± 0.1 | +0.6 | +8.9% |
> | Overall Average | 7.8 ± 0.1 | 7.3 ± 0.1 | +0.6 | +7.8% |
>
> ### Evaluation with DeepSeek-R1 as Judge:
>
> | Metric | Formal-CoT | Normal | Difference (F-N) | Improvement% |
> | --- | --- | --- | --- | --- |
> | Similarity Score | 6.5 ± 0.1 | 5.7 ± 0.1 | +0.8 | +12.6% |
> | Rationality Score | 9.0 ± 0.1 | 9.2 ± 0.1 | -0.1 | -1.3% |
> | Inclusiveness Score | 7.2 ± 0.1 | 6.6 ± 0.1 | +0.6 | +9.9% |
> | Overall Average | 7.6 ± 0.1 | 7.2 ± 0.0 | +0.4 | +5.9% |
>
> ### Key Findings:
>
> - **Most notably**, both evaluators consistently show remarkable improvements in similarity to expert answers (ranging from **12.6% to 14.1%**). This indicates that the formal reasoning process helps the model maintain a strong focus on the core problem, resulting in outputs that align much more closely with domain expertise in terms of key facts and logical structure.
> - Both evaluators also demonstrated substantial improvements in answer comprehensiveness (**8.9% to 9.9%**), further reinforcing the efficacy of the formal reasoning guidance.
> - Regarding the slight decrease in Rationality Score observed with DeepSeek evaluation (**1.3%**), it is important to note that both scores remain high (**9.0 vs 9.2**). This minor variation falls within the margin of error and likely reflects different calibration standards between evaluators rather than any degradation in reasoning quality.
> - The **overall average** **score improvements** (ranging from **5.9% to 7.8%**) across both evaluation settings provide robust and consistent evidence that our formal framework can significantly enhance model performance in specialized domains.
>
> ### Future Directions:
>
> This successful validation paves the way for three key applications we envision:
>
> - **As training data**: Future models can be trained to engage in "symbolic thinking," thus generating more reliable and interpretable natural language answers.
> - **As a verification module**: The framework offers the potential to automate the logical verification of scientific statements in a reproducible and transparent manner.
> - **As a quantifiable metric**: It converts subjective natural language evaluations into objective, structured assessments, providing more consistent and measurable evaluation criteria.
>
> We acknowledge that developing a comprehensive formal system for adjuvant research remains an ongoing challenge. The framework presented in this work establishes an extensible foundation for structured knowledge representation within this domain, and serves as an effective reasoning aid that substantially improves model performance. Moving forward, we are committed to expanding this work through the continued development of the formal dataset and further validation across various model architectures and task types.

---

> ### Author Response · Authors · 2025-11-19
>
> ## A (W4)：
>
> We thank the reviewer for raising the important concern about potential self-enhancement bias, which is a key consideration in studies using LLM-as-a-judge. To mitigate this risk, we implemented a two-pronged strategy. First, during the data generation phase, we used a diverse set of models (including GPT-4o, Claude3.5-Sonnet, Ernie4.0-Turbo, and DeepSeek-R1) to ensure that no single model’s style dominated the benchmark. For the generation process, each model was provided with **PDF documents** containing relevant content to generate questions. This ensured that the questions were based on the same structured input and not influenced by individual model biases. Second, during the evaluation phase, models were assessed in a strict zero-shot setting, with no prior exposure to the generated data or the benchmark, ensuring that the evaluation was independent of the generation process. Additionally, we employed two different models (GPT-4o and DeepSeek-R1) to perform the evaluation and assign scores to the answers. Our analysis shows that the relative rankings of the top models remained highly stable across both evaluators, further supporting the consistency and reliability of the results.
>
> However, we agree with the reviewer that these measures may not fully eliminate all biases, particularly those shared across models with similar reasoning styles. We acknowledge this as a limitation of our current approach. In the revised manuscript, we will explicitly acknowledge this potential for self-enhancement bias in the **Limitations** section. We also support the reviewer’s recommendation for future work to employ a fully independent panel of judge models that are entirely separate from the data generation process, to ensure more neutral and objective evaluation.
>
> ## A (W6)：
>
> We thank the reviewer for raising the issue of data release and licensing. To ensure the dataset's usability, we will fully open-source the benchmark following the paper's publication and provide clear licensing information to ensure the legal redistribution of images, textbook content, and literature excerpts.
>
> We have carefully addressed your questions and concerns. Should you have any further inquiries, please feel free to raise them, and we would be happy to provide additional clarifications and engage in further discussion. We sincerely hope that you will consider enhancing our score. Thank you very much for your thoughtful consideration.

---

> > ### Author Response · Authors · 2025-11-25
> >
> > Dear Reviewer,
> >
> > We would like to start by once again sincerely thanking you for the invaluable time and professional feedback you have provided for our manuscript. Your constructive comments have been instrumental in improving our work.
> >
> > We have completed our responses to your questions and submitted the supplementary explanations and experiments as part of the rebuttal process.
> >
> > Given that a period of time has elapsed since the rebuttal submission, we are writing to politely inquire about the current status of the further evaluation of our response. We fully appreciate and respect the professionalism and time required for the reviewing process.
> >
> > If further time is required for your deliberation, we completely understand and are willing to patiently await the final assessment.
> >
> > Thank you again for your diligent work and continued support. We look forward to receiving your valuable feedback.
> >
> > Sincerely

---

### Author Response · Authors · 2025-11-29
**Concern Regarding Paper Submission3923: Rebuttal Scores and Documented Reviewer Feedback**

Dear Area Chair,

We are the authors of the paper Submission3923.

We are aware that due to a technical issue with the visibility settings on OpenReview during the rebuttal period, the conference organizers have decided to revert all score changes resulting from the rebuttal. We understand and respect this decision made by the organizers, and we believe it is a necessary measure to uphold the academic integrity of the conference.

We would like to bring to your attention a key fact regarding our paper, and we kindly request that you take it into consideration during your final decision:

1. **Comprehensive and Diligent Responses**: During the rebuttal period (we submitted our final response between **19 Nov 2025, 20:40 and 19 Nov 2025, 21:10**), we **diligently addressed every comment (Weaknesses and questions) raised by all reviewers.**

2. **Effective and Substantive Interaction**: Our responses successfully persuaded two of the reviewers, who engaged with us in **positive and substantive discussions** and **explicitly stated their intention to increase their scores based on our replies**. Importantly, these interactions occurred before the OpenReview Bug was widely reported (27 Nov 2025):

- **Reviewer 2Z13** responded on **20 Nov 2025, 03:01**, stating: **"these results look good. I revised my score accordingly. good luck."**

- **Reviewer 2T68**: It is particularly noteworthy that after we sent **a polite follow-up inquiry on 25 Nov 2025, 19:45**, the reviewer subsequently responded on **26 Nov 2025, 20:40**, explicitly stating: **"experiments on both possible contamination with LLM as a judge and using the formal description look convincing. I am increasing my score, good luck."** This sequence confirms that the reviewer's positive assessment and decision to increase the score were a direct and considered response to our rebuttal.

These clearly documented discussions, which took place prior to the technical issue, demonstrate that the rebuttal process was effective and had a positive impact on the assessment of our work. Simply reverting the scores without acknowledging these successful academic exchanges would, in our case, be inequitable.

Therefore, we earnestly request that you consider this **successful rebuttal discussion as a significant factor** in your final deliberation. We are not asking for the system to forcibly revert the scores, but rather we wish to ensure you are aware that the initial scores from these two reviewers do not fully reflect their final, positive opinion formed after reading our rebuttal.

We fully understand that the conference organizers have entrusted the final authority on this matter to the Area Chairs. We sincerely appreciate you taking the time to review our situation amidst your busy schedule and we truly recognize the heavy workload and effort involved in your role. Thank you very much for your time and dedication.

Sincerely

---

### Meta-Review · Area_Chair_FqJB · 2026-01-07

**Summary:**

This paper introduces a domain-specific benchmark and structured knowledge resource for evaluating LLMs/MLLMs in vaccine adjuvant research. The authors evaluate 29 models using embedding-based metrics and LLM-as-a-judge rubrics, and propose a formalized representation aimed at supporting future neuro-symbolic reasoning and domain-specialized systems.

Reviewers raised largely consistent concerns regarding (i) potential self-enhancement bias when models are used both for data generation and as LLM judges, (ii) the role of the formal description and whether it provides demonstrated downstream benefit, and (iii) limited dataset size and relatively weak statistical analysis for some results.

The authors responded clearly in rebuttal, providing additional ablation studies, alternative judge analyses, and preliminary experiments to support the utility of the formal representation, while also acknowledging remaining limitations. Although some methodological concerns remain—particularly regarding the clarity of the multimodal evaluation setup—this work represents a meaningful first step toward a much-needed benchmark in an underexplored domain and received majority support for acceptance from the reviewers. Overall, I lean toward acceptance, with the expectation that the final version will more explicitly state limitations and improve transparency of experimental settings.

**Reviewer Concerns:**

Addressed concerns
- Potential self-enhancement bias when models are used both for data generation and as LLM judges. This concern was substantially addressed in the rebuttal. The authors provided additional ablation studies, evaluated models on subsets excluding their own generated data, and introduced alternative, independent judge models.
- Role of the formal description and demonstrated downstream benefit. The authors clarified the intended role of the formal representation and presented preliminary controlled experiments showing that incorporating formal language guidance can improve answer quality.
- Relatively weak statistical analysis. The authors supplemented the rebuttal with additional analyses.

Outstanding concerns:
- Potentially misleading use of “multimodal” and reliance on OCR. The concern raised by reviewer Z32D regarding the characterization of the benchmark as multimodal remains outstanding. Although the authors stated an intention to revise terminology and clarify limitations, these changes are not yet reflected in the current draft. The AC agrees with the reviewer that explicit acknowledgment that OCR-based processing reduces genuine multimodal reasoning, is necessary in the final version.
- Reproducibility and data licensing clarity. While the authors promised to release full settings and licensing details alongside the benchmark, these details are not yet clearly specified in the paper. Explicit reporting of experimental parameters and data licensing are still required to ensure reproducibility and usability.
- Limited dataset size for certain subsets. The authors acknowledge this and frame the work as an initial benchmark.

**Reviewer Scores:**

- Reviewer Z32D (4): This reviewer raised concerns about the misleading use of the term “multimodal,” reliance on OCR, reproducibility gaps, and licensing clarity. While some of these issues were acknowledged in the rebuttal, they are not yet fully addressed in the current draft. I therefore expect this reviewer would likely keep their original score.
- Reviewer 2Z13, 2T68, zNQ7 (6): I expect their score to remain unchanged.

Overall, a majority of reviewers lean toward acceptance. While some concerns remain outstanding, particularly around multimodal framing and reproducibility, the consensus after discussion trends in favor of acceptance.

---

### Decision · Program_Chairs · 2026-01-26

Accept (Poster)